# Thompson Sampling for Multinomial Logit Contextual Bandits

**Min-hwan Oh**
Columbia University
New York, NY
m.oh@columbia.edu

**Garud Iyengar**
Columbia University
New York, NY
garud@ieor.columbia.edu

## Abstract

We consider a dynamic assortment selection problem where the goal is to offer a sequence of assortments that maximizes the expected cumulative revenue, or alternatively, minimize the expected regret. The feedback here is the item that the user picks from the assortment. The distinguishing feature in this work is that this feedback is given by a multinomial logit choice model. The utility of each item is a dynamic function of contextual information of both the item and the user. We refer to this problem as the multinomial logit contextual bandit. We propose two Thompson sampling algorithms for this multinomial logit contextual bandit. Our first algorithm maintains a posterior distribution of the unknown parameter and establishes $\widetilde{\mathcal{O}}(d\sqrt{T})$[1] Bayesian regret over $T$ rounds with $d$ dimensional context vector. The second algorithm approximates the posterior by a Gaussian distribution and uses a new optimistic sampling procedure to address the issues that arise in worst-case regret analysis. This algorithm achieves $\widetilde{\mathcal{O}}(d^{3/2}\sqrt{T})$ worst-case (frequentist) regret bound. The numerical experiments show that the practical performance of both methods is in line with the theoretical guarantees.

## 1 Introduction

In the stochastic multi-armed bandit (MAB) problem [10, 27], the learning agent selects one of $N$ actions (or items) and receives a revenue feedback corresponding to the chosen action in each round. The objective is to maximize the cumulative revenue over a finite horizon of length $T$, or alternatively, to minimize the cumulative regret defined as the difference in cumulative revenues of the optimal strategy and the agent's strategy. The main challenge in MAB problems is to appropriately balance the trade-off between exploitation, i.e., pulling the best empirical arm, and exploration, i.e., experimenting with arms which are not sufficiently pulled. The balancing strategies for this exploration-exploitation trade-off typically fall into two categories: upper confidence bound (UCB) methods [9, 18] and Thompson sampling (TS) based methods [42]. (Besides UCB and TS, one may also consider $\epsilon$-greedy approach [24].)

UCB methods maintain a confidence set for the unknown true parameter, and in each step, choose the most optimistic parameter from this set, and pull the optimal arm corresponding to this optimistic parameter value. The confidence set is updated based on the revenue feedback which is revealed after an arm is pulled. TS assumes a prior distribution over the parameters defining the reward distribution. At each step, a parameter value is sampled from the posterior distribution, and an optimal arm corresponding to a sampled parameter is pulled. Upon observing the reward for each round, the posterior distribution is updated via Bayes rule. TS has been successfully applied in a wide range of settings [40, 13, 38].

While UCB algorithms have simple implementations and good theoretical regret bounds [29], TS has been shown to achieve better empirical performance in many simulated and real-world settings without sacrificing simplicity [13, 23]. In order to bridge this gap, many recent studies have been focused on the analysis of worst-case regret and Bayesian regret in TS approaches for both contextual bandits and reinforcement learning settings [5, 7, 38, 3]. The main technical difficulty in analyzing regret in the TS lies in controlling the deviation introduced by the randomness in the algorithm.

In this paper, we consider a dynamic assortment selection with contextual information, which is a combinatorial variant of the contextual bandit problem. The goal is to offer a sequence of assortments of at most $K$ items from a set of $N$ possible items that minimize regret. The feedback here is the particular item chosen by the user from the offered assortment. This problem arises in many real-world applications such as online retailing, streaming services, news feed, online advertising, etc. We assume that the item choice is given by a multinomial logit (MNL) choice model [33]. This is one of the most widely used models in dynamic assortment optimization literature [12, 37, 39, 6, 7, 14]. The utility of each item that defines the MNL choice probability is assumed to be a linear function of a $d$-dimensional contextual information, or a set of $d$ features. This contextual information can be a combined information of *both* the item and the user, and is allowed to change over time.

The MNL contextual bandit is a multinomial generalization of generalized linear contextual bandits [23, 30], particularly logistic bandits, that reduces to generalized linear bandits when the assortment contains a single item. However, this extension is non-trivial since the MNL model cannot be expressed in the form of a generalized linear model [15]; hence, the results of generalized linear bandits do not directly apply. Also, in contrast to the standard contextual bandit problems, in the MNL contextual bandit, the item choice (feedback) is a function of the entire offered assortment. Thus, regret analysis is more complicated. Furthermore, we allow the context vector to vary arbitrarily in time; thus, offering the same assortment repeatedly several times to learn the parameter values [6, 7] is no longer an effective strategy.

We propose two Thompson sampling algorithms for this multinomial logit contextual bandit. To our knowledge, these are the first TS algorithms for this problem.

(a) The first algorithm maintains a posterior distribution of the true parameter and establishes $\widetilde{\mathcal{O}}(d\sqrt{T})$ Bayesian regret.

(b) The second algorithm approximates the posterior by a Gaussian distribution and uses a new optimistic sampling procedure to address the issues that arise in worst-case regret analysis. We establish $\widetilde{\mathcal{O}}(d^{3/2}\sqrt{T})$ worst-case (frequentist) regret bound for this algorithm.

The additional $\sqrt{d}$ factor in the regret of the second algorithm is due to the deviation from the random sampling in TS which is addressed in the worst-case regret analysis and is consistent with the results in TS methods for linear bandits [5, 3]. Both regret bounds are free of candidate item set size $N$, which implies that our TS algorithms can be applied to a large item set. The TS algorithms we propose are efficient to implement as long as the assortment optimization step is solved efficiently, for which our TS algorithms can exploit efficient polynomial-time algorithms [36, 20], which is a significant advantage over the previously proposed UCB method in [15] which computes the confidence bound for *each assortment* (i.e., for each of the total $N$ choose $K$ assortments). Furthermore, the numerical experiments show that the practical performance of the proposed methods is in line with the theoretical guarantees.

## 2   Related Work

The MNL model [34, 33, 32] is one of the most widely used choice models for assortment selection problems. The problem of computing the optimal assortment (*static* assortment optimization problem), when the MNL parameters, i.e., user preferences, are known a priori, is well-studied [41, 21, 22]. Our work belongs to the literature on *dynamic* assortment optimization. [12] consider the setting where the demand for items in an assortment is independent. [37] and [39] consider the problem of minimizing regret under the MNL choice model and present an "explore first then exploit later" approach. [37] showed $\mathcal{O}(N^2 \log^2 T)$ regret bound, where $N$ is the number of total candidate items. [39] later improved the bound to $\mathcal{O}(N \log T)$. However, these methods require a priori knowledge of "separability" between the true optimal assortment and the other sub-optimal alternatives.

More recent work by [6, 7, 16, 14, 15] also incorporated MNL models into dynamic assortment optimization and formulated the problem into an online regret minimization problem without requiring a priori knowledge on separability. [6] proposed UCB-style algorithm which shows $\widetilde{\mathcal{O}}(\sqrt{NT})$ regret bound. [7] achieve the same order of the regret bound $\widetilde{\mathcal{O}}(\sqrt{NT})$ using TS approach with improved empirical performance. [14] show a matching lower bound of $\Omega(\sqrt{NT})$. All of this previous work on MNL bandits assumes each item is associated with a unique parameter, i.e., one cannot learn across items. In our proposed MNL contextual bandits, the utility of item $i$ at round $t$ is of the form $x_{ti}^\top \theta^*$ some fixed but unknown *utility parameter* $\theta^*$; hence, we can learn across items. When the feature dimension $d \ll \sqrt{N}$, learning across items allows one to reduce the regret bound from $\widetilde{\mathcal{O}}(\sqrt{NT})$ to $\widetilde{\mathcal{O}}(d\sqrt{T})$. However, one cannot directly incorporate (time-varying) contextual information into the previous work (see, e.g. [6, 7]) since these methods require that the same assortment be offered repeatedly for a random number of rounds until an outside choice (no purchase) is observed. [15] proposed a UCB method which establishes $\widetilde{\mathcal{O}}(d\sqrt{T})$ regret bound for the MNL contextual bandit similar to our settings. Apart from the fact that their method is UCB based, there is another fundamental difference between [15] and our work. [15] enumerates the exponentially many ($N$ choose $K$) assortments and builds confidence bounds for each of them. In contrast, our methods only maintain uncertainty for each of the $N$ different items.

It is also worth mentioning work in the personalized MNL-bandit problem [25, 17, 11]. These works consider each item utility separately and learn $N$ different parameters; hence there is no generalization across different items, which is different from our setting. Perhaps, the most related one among these personalized MNL bandit methods is [17], which proposed a TS algorithm for their problem. However, they only provide the Bayesian regret which is relatively easier to control compared to the worst-case regret (we discuss this aspect in Section 5), and again their method (as well as other personalized MNL bandit methods) still considers learning $N$ separate parameters for each of the items; hence it is not scalable for a large item set (i.e., large $N$).

Linear contextual bandits [2, 9, 19, 36, 1, 18, 5] have been widely studied. [23] and [30] extend the linear contextual bandit to scalar, monotone, generalized linear bandit using a UCB-type approach. In most of these linear bandits or generalized linear bandits, balancing exploitation and exploration can be done simply by taking an action that maximizes the sum of mean reward and the variance. [5] define TS for linear contextual bandit as a Bayesian algorithm where a Gaussian prior over $\theta^*$ is updated according to the observed rewards, a random sample is drawn from the posterior, and the corresponding optimal arm is selected at each step. They show $\widetilde{\mathcal{O}}(d^{3/2}\sqrt{T})$ worst-case regret bound. Following the work of [5], [3] show that the TS does not need to sample from an actual Bayesian posterior distribution and that any distribution satisfying suitable concentration and anti-concentration properties guarantees a small regret and provide an alternative proof of TS achieving the same regret bound $\widetilde{\mathcal{O}}(d^{3/2}\sqrt{T})$. However, these results in (generalized) linear contextual bandits (either UCB or TS) do not apply directly to our MNL contextual bandit problem, since the choice probability of an item in an assortment is non-linear and non-monotone in the MNL parameter $\theta^*$. It is also worthwhile to mention a line of work in other combinatorial bandit problems [35, 43, 26] mostly with semi-bandit feedback or cascading feedback. Our work is distinct from these combinatorial bandit problems since in cascading or semi-bandit settings, the mapping from the item context to the user feedback is still independent of other items in an offered set; hence it does not take substitution effect into account. On the other hand, MNL choice feedback is a function of the entire assortment which makes our analysis more challenging.

## 3 Problem Formulation

### 3.1 Notations

For a vector $x \in \mathbb{R}^d$, we use $\|x\|$ to denote its $\ell_2$-norm and $x^\top$ its transpose. The weighted $\ell_2$-norm associated with a positive-definite matrix $V$ is defined by $\|x\|_V := \sqrt{x^\top V x}$. The minimum and maximum singular values of a matrix $V$ are written as $\lambda_{\min}(V)$ and $\|V\|$, respectively. The trace of a matrix $V$ is trace($V$). For two symmetric matrices $V$ and $W$ of the same dimensions, $V \succeq W$ means that $V - W$ is positive semi-definite. We define $[n]$ for a positive integer $n$ to be a set containing positive integers up to $n$, i.e., $\{1, 2, ..., n\}$. Finally, we define $\mathcal{S}$ to be the set of candidate assortments with size constraint at most $K$, i.e., $\mathcal{S} = \{S \subset [N] : |S| \leq K\}$.

## 3.2 MNL Contextual Bandits

We formulate the problem of the MNL contextual bandit as follows. The decision-making agent can choose an assortment as a subset of the item set containing $N$ distinct items, indexed by $i \in [N]$. At round $t$, feature vectors $x_{ti} \in \mathbb{R}^d$ for every item $i \in [N]$ are revealed to the agent. Each feature vector combines the information of the user and the corresponding item $i$. For example, suppose the user at round $t$ is characterized by a feature vector $v_t$ and the item $i$ has a feature vector $w_{ti}$ (note that we allow feature vectors for an item and a user to change over time), then we can use $x_{ti} = \text{vec}(v_t w_{ti}^\top)$, the vectorized outer-product of $v_t$ and $w_{ti}$, as the combined feature vector of item $i$ a at round $t$. If $v_t$ is not available, we can use item dependent features only $x_{ti} = w_{ti}$. Given this contextual information, at every round $t$, the agent selects an assortment $S_t \in \mathcal{S}$ and observes the user choice represented as a binary vector $y_t \in \{0, 1\}^{|S_t|}$ where $y_{ti} = 1$ if the $i$-th item in assortment $S_t$ is chosen by the user and $y_{tj} = 0$ for all non-chosen items $j \in S_t$. Note that $\sum_{i \in S_t} y_{ti} \leq 1$ and we allow an "outside option" ($i = 0$) which means the user does not choose any items offered in $S_t$, i.e., $y_{ti} = 0$ for all $i \in S_t$. This user choice is given by the MNL choice model. Under this model, the probability that a user chooses item $i \in S_t$ is given by,

$$p_{ti}(S_t, \theta^*) = \frac{\exp\{x_{ti}^\top \theta^*\}}{1 + \sum_{j \in S_t} \exp\{x_{tj}^\top \theta^*\}}$$

where $\theta^* \in \mathbb{R}^d$ is an unknown time-invariant parameter and 1 in the denominator accounts for the outside option with $p_{t0}(S_t, \theta^*) = 1/(1 + \sum_{j \in S_t} \exp\{x_{tj}^\top \theta^*\})$. Then, the choice response variable $y_t = (y_{t0}, y_{t1}, ..., y_{tK})$ is a sample from this multinomial distribution:

$$y_t \sim \text{multinomial}\big(1, p_{t0}(S_t, \theta^*), p_{t1}(S_t, \theta^*), ..., p_{tK}(S_t, \theta^*)\big)$$

where 1 represents $y_t$ is a single-trial sample. Also, we define noise $\epsilon_{ti} := y_{ti} - p_{ti}(S_t, \theta^*)$. Since $\epsilon_{ti}$ is bounded in $[0, 1]$, $\epsilon_{ti}$ is $\sigma^2$-sub-Gaussian with $\sigma^2 = 1/4$. It is important to note that $\epsilon_{ti}$ is not independent across $i \in S_t$ due to the substitution effect in the MNL model.

The revenue parameter for each item $i$ is also revealed at round $t$, denoted by $r_{ti}$. Note that $r_{ti}$ is the revenue incurred by item $i$ if item $i$ is chosen by the user at round $t$. Without loss of generality, we assume $|r_{ti}| \leq 1$ for all $i$ and $t$. Then, the expected revenue corresponding to assortment $S_t$ is given by

$$R_t(S_t, \theta^*) = \sum_{i \in S_t} \frac{r_{ti} \exp\{x_{ti}^\top \theta^*\}}{1 + \sum_{j \in S_t} \exp\{x_{tj}^\top \theta^*\}}.$$

Let $S_t^*$ be the offline optimal assortment at round $t$ under full information when $\theta^*$ is known, i.e., if the true MNL probabilities $p_{ti}(S, \theta^*)$ are known a priori:

$$S_t^* = \arg\max_{S \in \mathcal{S}} R_t(S, \theta^*).$$

Consider a planning horizon $T$, where assortments can be offered at rounds $t = 1, ..., T$. The agent does not know the value of $\theta^*$ (hence $p_{ti}(S, \theta^*)$ is not known) and can only make sequential assortment decisions, $S_1, ..., S_T$ at rounds $1, ..., T$ respectively. Hence, the main challenge is how to construct an algorithm that simultaneously learns the unknown parameter $\theta^*$ and sequentially makes the decisions on offered assortments based on past choices and observed responses to maximize cumulative expected revenues over the planning horizon. The performance of an algorithm is usually measured by the regret, which is the gap between the expected revenue generated by the assortment chosen by the algorithm and that of the offline optimal assortment. We define the (worst-case) cumulative expected regret as

$$\mathcal{R}(T, \theta^*) = \sum_{t=1}^{T} \mathbb{E}\big[R_t(S_t^*, \theta^*) - R_t(S_t, \theta^*) \mid \theta^*\big]$$

where $R_t(S_t^*, \theta^*)$ is the expected revenue corresponding to the offline optimal assortment at round $t$, and the expectation is taken over random parameters and possible randomization in a learning algorithm. When it is clear that we condition on a fixed $\theta^*$, we denote $\mathcal{R}(T) := \mathcal{R}(T, \theta^*)$ in the rest of the paper. In Bayesian settings, i.e., when $\theta^*$ is randomly generated or the learning agent has a prior belief in $\theta^*$, the Bayesian cumulative regret [38] over $T$ horizon is defined as

$$\mathcal{R}_{\text{Bayes}}(T) = \mathbb{E}_{\theta^*}[\mathcal{R}(T, \theta^*)] = \sum_{t=1}^{T} \mathbb{E}\big[R_t(S_t^*, \theta^*) - R_t(S_t, \theta^*)\big]$$

where the expectation is taken also over the distribution of $\theta^*$. In other words, $\mathcal{R}_{\text{Bayes}}(T)$ is a weighted average of $\mathcal{R}(T, \theta^*)$ under the prior on $\theta^*$.

### 3.3 Assumptions

We introduce general assumptions on the structure of the problem.

**Assumption 1.** $\|x_{ti}\| \leq 1$ *for all $t$ and $i$. Also, $\|\theta^*\| \leq 1$.*

This assumption is used to make the regret bounds scale-free for convenience and is in fact standard in the bandit literature. If $\|x_{ti}\| \leq C$ and $\|\theta^*\| \leq C$ for some constant $C$ instead, then our regret bounds would increase by a factor of $C$.

**Assumption 2.** *There exists $\kappa > 0$ such that for every item $i \in S$ and any $S \in \mathcal{S}$ and all round $t$* $\inf_{S \in \mathcal{S}, \theta \in \mathbb{R}^d} p_{ti}(S, \theta) p_{t0}(S, \theta) \geq \kappa.$

Note that this is equivalent to a standard assumption in generalized linear contextual bandit literature [23, 30] to ensure the Fisher information matrix is invertible and is adapted to suit our MNL setting. We discuss the need for this assumption in detail in Appendix A.

## 4   Algorithm: TS-MNL

In this section, we describe TS-MNL, our first TS algorithm for the MNL contextual bandit problem, and present its Bayesian regret bound. We first provide the definition of the posterior distribution $Q_t$ on the unknown parameter $\theta^*$. At the beginning of the learning phase, the agent knows that $\theta^*$ is distributed according to $Q_0$, the prior distribution. Now, at each round $t$, the agent has access to the observations up to round $t$, $\mathcal{D}_t = \{X_\tau, y_\tau\}_{\tau=1}^{t-1}$ where $X_\tau = \{x_{\tau i}\}_{i \in S_\tau}$. Then the agent combines $Q_0$ and $\mathcal{D}_t$ to define the posterior distribution $Q_t(\theta)$:

$$Q_t(\theta) \propto Q_0(\theta) p(\mathcal{D}_t | \theta), \quad \text{where } p(\mathcal{D}_t | \theta) = \prod_{\tau=1}^{t-1} \prod_{i \in S_\tau} (p_{\tau i}(S_\tau, \theta))^{y_{\tau i}} \tag{1}$$

and the "$\propto$" notation hides the partition function $\int_\phi Q_0(\phi) p(\mathcal{D}_t | \phi) d\phi$ in the denominator. In other words, the posterior distribution is proportional to the product of the prior distribution and the likelihood function. Note that there is no conjugate prior for the MNL model. Hence, sampling from $Q_t$ is intractable. In order to overcome this intractability, one may draw an approximate sampling using Markov chain Monte Carlo [8]. For ease of exposition, we assume the following in this section and in the Bayesian regret analysis. We will later provide a remedy for this intractability in the modification of our algorithm for the worst-case regret analysis.

**Assumption 3.** *We can sample from $Q_t(\theta)$.*

In each round $t$, TS-MNL algorithm consists of three major steps. First, it randomly samples a parameter $\widetilde{\theta}_t$ from the posterior distribution $Q_t$. Second, it computes the assortment choice $S_t$ under this sampled parameter $\widetilde{\theta}_t$. Finally, $S_t$ is offered to the user and feedback $y_t$ is observed. The pseudocode of TS-MNL is presented in Algorithm 1.

---

**Algorithm 1** TS-MNL

---

1: **Input**: prior distribution $Q_0$
2: **for** all $t = 1$ to $T$ **do**
3:     Observe $x_{ti}$ and $r_{ti}$ for all $i \in [N]$
4:     Sample $\widetilde{\theta}_t$ from the posterior distribution $Q_t$ in Eq.(1)
5:     Compute $S_t = \arg\max_{S \in \mathcal{S}} R_t(S, \widetilde{\theta}_t)$
6:     Offer $S_t$ and observe $y_t$ (user choice at round $t$)
7: **end for**

---

**Combinatorial Optimization.** Algorithm 1 has the combinatorial optimization step in Line 5. There are efficient polynomial-time algorithms available to solve this combinatorial optimization problem [37, 20] for given utility estimates under the sampled parameter. In particular, we can use the solution of the linear programming (LP) formulation presented in [20] for this optimization step.

## 4.1 Bayesian Regret of TS-MNL

We state the Bayesian cumulative regret bound for Algorithm 1 in Theorem 1. We also provide an overview of establishing the regret bound.

**Theorem 1.** *Suppose we run* TS-MNL *(Algorithm 1) for a total of $T$ rounds with assortment size constraint $K$. Then the Bayesian regret of the algorithm is upper-bounded by*

$$\mathcal{R}_{Bayes}(T) \leq \mathcal{O}(1) + \left[\frac{1}{\kappa}\sqrt{2d\log\left(1 + \frac{TK}{d^2}\right) + 2\log T} + \frac{\sqrt{d}}{\kappa}\right] \cdot \sqrt{2dT\log\left(1 + \frac{TK}{d^2}\right)}$$

$$= \mathcal{O}\left(d\sqrt{T}\log\left(1 + \frac{TK}{d^2}\right)\right).$$

Theorem 1 establishes $\widetilde{\mathcal{O}}(d\sqrt{T})$ Bayesian regret. [15] established the lower bound $\Omega(d\sqrt{T}/K)$ for MNL contextual bandits under almost identical settings. When $K$ is small and fixed (which is typically true in many applications), Theorem 1 demonstrates that TS-MNL is almost optimal. Furthermore, the regret bound is completely free of $N$; hence TS-MNL is applicable to the case of a large number of items (large $N$). Also, if $K \leq d^2$, the regret bound becomes free of $K$. In Section 6, we introduce modifications to TS-MNL for the worst-case regret analysis which include the explicit use of regularized MLE for parameter estimation and sampling from the Gaussian distribution instead of maintaining the actual posterior to overcome the intractability. The concentration results derived for the Bayesian regret analysis in this section serve as a building block for the worst-case regret analysis for the modified algorithm.

The proof outline of Theorem 1 is motivated by [38, 43]. Given $\mathcal{F}_t$ which contains all available information up to round $t$, $\widetilde{\theta}_t$ and $\theta^*$ are i.i.d. with the posterior distribution $Q_t$ in the Bayesian perspective. Also, the optimization step is a fixed combinatorial optimization and $\{x_{ti}\}_{i\in[N]}$ are fixed given $\mathcal{F}_t$. Hence, conditioning on $\mathcal{F}_t$, $S_t$ and $S_t^*$ are also i.i.d. Therefore, the expected regret pertaining to the random sampling is 0.; Then, we control the estimation error of $\theta^*$ for which we utilize the finite-sample concentration results for MNL parameter. The proofs are left to Appendix B.

## 5 Worst-Case Regret

Algorithm 1 is still valid under a frequentist setting, i.e., when the true parameter is not a random variable but a fixed parameter. However, when analyzing the worst-case regret (also known as frequentist regret) for the algorithm, the main technical difficulty lies in controlling the deviation in performance due to the random sampling of the algorithm. Note that in Bayesian regret analysis, controlling this sampling deviation is not addressed because of the assumption that $\widetilde{\theta}_t$ and $\theta^*$ are i.i.d. conditioning on $\mathcal{F}_t$. However, this does not hold anymore when $\theta^*$ is fixed; hence the worst-case regret analysis needs to ensure that the deviation due to sampling is small enough. To see this, we decompose the worst-case immediate regret into a few components.

$$\mathcal{R}(t) = \mathbb{E}[R_t(S_t^*, \theta^*) - R_t(S_t, \theta^*)]$$
$$= \mathbb{E}[R_t(S_t^*, \theta^*) - R_t(S_t^*, \widetilde{\theta}_t)] + \mathbb{E}[R_t(S_t^*, \widetilde{\theta}_t) - R_t(S_t, \widetilde{\theta}_t)] + \mathbb{E}[R_t(S_t, \widetilde{\theta}_t) - R_t(S_t, \theta^*)]$$
$$\leq \mathbb{E}[R_t(S_t^*, \theta^*) - R_t(S_t^*, \widetilde{\theta}_t)] + \mathbb{E}[R_t(S_t, \widetilde{\theta}_t) - R_t(S_t, \theta^*)] \quad (2)$$

The inequality comes from the fact that our assortment choice at round $t$, $S_t$, is optimal under $\widetilde{\theta}_t$; hence $R_t(S_t^*, \widetilde{\theta}_t) \leq R_t(S_t, \widetilde{\theta}_t)$. The second term $\mathbb{E}[R_t(S_t, \widetilde{\theta}_t) - R_t(S_t, \theta^*)]$ in Eq.(2) is relatively easier to control. We can show that the term can be bounded by combining the upper-bound for the estimation error $|x^\top(\hat{\theta}_t - \theta^*)|$ and the concentration of the sampling probability of $\widetilde{\theta}_t$. However, controlling the first term $\mathbb{E}[R_t(S_t^*, \theta^*) - R_t(S_t^*, \widetilde{\theta}_t)]$ in Eq.(2) is more challenging in frequentist analysis. First, note that $\mathbb{E}[R_t(S_t^*, \theta^*) - R_t(S_t^*, \widetilde{\theta}_t)] = 0$ in the Bayesian regret by the assumption that $\theta^*$ and $\hat{\theta}_t$ are i.i.d. conditioning on $\mathcal{F}_t$ as mentioned earlier. However, this is no longer true in the worst-case regret analysis. In the worst-case regret analysis of TS, this term is controlled by showing that a sampled parameter is optimistic frequently enough. In other words, we need to lower-bound the probability of the sampled parameter being optimistic, i.e., $\mathbb{P}(R_t(S_t^*, \widetilde{\theta}_t) \geq R_t(S_t^*, \theta^*) \mid \mathcal{F}_t) \geq p$ for some parameter free $p > 0$.

To describe the challenge in our MNL contextual bandit problem, we present the following lemma which shows that the expected revenue for the optimal assortment is monotonically increasing with an increase in the utility estimates.

**Lemma 1** ([6], Lemma 4.2). *Suppose $S_t^*$ is the optimal assortment under the true parameter $\theta^*$ at round $t$, i.e., $S_t^* = \arg\max_{S \in \mathcal{S}} R_t(S, \theta^*)$. Also suppose that $x_{ti}^\top \theta^* \leq x_{ti}^\top \theta'$ for all $i \in S_t^*$. Then $R_t(S_t^*, \theta^*) \leq R_t(S_t^*, \theta')$.*

Note that Lemma 1 shows the monotonicity of expected revenue only for the optimal assortment and it does not claim that the expected revenue is generally a monotone function for all assortments. This lemma implies that we can lower-bound the probability of having an optimistic expected revenue under the sampled parameter.

$$\mathbb{P}\left(R_t(S_t^*, \widetilde{\theta}_t) \geq R_t(S_t^*, \theta^*) \mid \mathcal{F}_t\right) \geq \mathbb{P}\left(x_{ti}^\top \widetilde{\theta}_t \geq x_{ti}^\top \theta^*, \forall i \in S_t^* \mid \mathcal{F}_t\right)$$

However, this makes the probability of being optimistic exponentially small in the size of the assortment $S_t^*$, i.e., exponentially small in $\mathcal{O}(K)$, which in turn results in exponential dependence on $\mathcal{O}(K)$ in the worst-case regret bound. In order to overcome such an issue, we adopt a few modifications in the algorithm which we discuss in the following section.

## 6 TS-MNL with Optimistic Sampling

**Sampling from Gaussian Distribution.** We modify our TS algorithm to a generic randomized algorithm constructed on the regularized MLE rather than sampling from an actual Bayesian posterior. [3] show that TS does not need to sample from an actual posterior distribution and that any distribution satisfying suitable concentration and anti-concentration properties guarantees a small regret. Specifically, instead of sampling from the posterior $Q_t$, we sample $\widetilde{\theta}_t$ from Gaussian distribution $\mathcal{N}(\hat{\theta}_t, \alpha_t^2 V_t^{-1})$ where $\hat{\theta}_t$ is the regularized MLE, the minimizer of Eq.(3), and $\alpha_t$ is the confidence radius. This way, we ensure tractability of the sampling distribution. Furthermore, this Gaussian approximation allows us to adopt optimistic sampling (which we discuss below) in an efficient manner.

**Optimistic Sampling.** The optimistic sampling we present here is a key ingredient in avoiding the theoretical challenges present in the worst-case regret analysis. For optimistic sampling, instead of drawing a single sample $\widetilde{\theta}_t$, we draw $M$ independent samples $\{\widetilde{\theta}_t^{(j)}\}_{j=1}^M$ from $\mathcal{N}(\hat{\theta}_t, \alpha_t^2 V_t^{-1})$ (the exact value of $M$ is specified in Theorem 2). Then we compute the optimistic utility estimate $\widetilde{u}_{ti}$ for each $i \in [N]$:

$$\widetilde{u}_{ti} = \max_j x_{ti}^\top \widetilde{\theta}_t^{(j)}.$$

We define $\widetilde{R}_t(S)$ to be the expected revenue of assortment $S$ based on $\widetilde{u}_{ti}$:

$$\widetilde{R}_t(S) = \frac{\sum_{i \in S} r_{ti} \exp\{\widetilde{u}_{ti}\}}{1 + \sum_{j \in S} \exp\{\widetilde{u}_{tj}\}}$$

Note that this optimistic sampling scheme is different from that proposed in [7]. The setting in [7] is non-contextual, and they use a 1-dimensional Gaussian random variable to correlate the samples of the utility of the $K$ items in order to ensure the probability that all samples are simultaneously optimistic is a constant. This correlated sampling reduces the overall variance severely, hence they propose taking $K$ samples instead of a single sample to increase the variance. In contrast, we take multiple samples of the multivariate Gaussian distribution to directly ensure that the probability of an optimistic sample is sufficiently large.

The pseudocode of the modified algorithm is presented in Algorithm 2. As before, we can utilize the LP solution [20] for the optimization step in Line 6. The modified algorithm now explicitly maintains the matrix $V_t$ and computes the regularized MLE $\hat{\theta}_t$. Note that $\alpha_T$ can be replaced by $\alpha_t = \mathcal{O}\left(\sqrt{d \log\left(1 + \frac{tK}{d\lambda}\right) + 4 \log t}\right)$ at round $t$, if the planning horizon $T$ is not known and the analysis holds for either case.

---

**Algorithm 2** TS-MNL with Optimistic Sampling

---

1: **Input**: sample size $M$, confidence radius $\alpha_T$, penalty parameter $\lambda$
2: **for** all $t = 1$ to $T$ **do**
3:     Observe $x_{ti}$ and $r_{ti}$ for all $i \in [N]$
4:     Sample $\{\widetilde{\theta}_t^{(j)}\}_{j=1}^M$ independently from $\mathcal{N}(\hat{\theta}_t, \alpha_T^2 V_t^{-1})$
5:     Compute $\widetilde{u}_{ti} = \max_j x_{ti}^\top \widetilde{\theta}_t^{(j)}$ for all $i \in [N]$
6:     Compute $S_t = \arg \max_{S \in \mathcal{S}} \widetilde{R}_t(S)$
7:     Offer $S_t$ and observe $y_t$ (user choice at round $t$)
8:     Update $V_{t+1} \leftarrow V_t + \sum_{i \in S_t} x_{ti} x_{ti}^\top$
9:     Compute the regularized MLE $\hat{\theta}_t$ by minimizing

$$-\sum_{\tau=1}^t \sum_{i \in S_\tau} y_{\tau i} \log p_{\tau i}(S_\tau, \theta) + \frac{\lambda}{2} \|\theta\|^2. \tag{3}$$

10: **end for**

---

## 6.1 Worst-Case Regret of TS-MNL with Optimistic Sampling

**Theorem 2.** *Suppose we run* TS-MNL *with "optimistic sampling" (Algorithm 2) for a total of $T$ rounds with optimistic sample size $M = \lceil 1 - \frac{\log K}{\log(1 - 1/(4\sqrt{e\pi}))} \rceil$, the penalty parameter $\lambda \geq 1$ and assortment size constraint $K$. Then the worst-case regret of the algorithm is upper-bounded by*

$$\mathcal{R}(T) \leq \mathcal{O}(1) + 16\sqrt{e\pi}\beta_T \left( \sqrt{2dT \log\left(1 + \frac{TK}{d\lambda}\right)} + \sqrt{\frac{8T}{\lambda} \log 2T} \right)$$

$$+ (\alpha_T + \beta_T)\sqrt{2dT \log\left(1 + \frac{TK}{d\lambda}\right)}$$

*where $\alpha_T = \frac{1}{2\kappa}\sqrt{d \log\left(1 + \frac{TK}{d\lambda}\right) + 4\log T} + \frac{\sqrt{\lambda}}{\kappa}$ and $\beta_T = \alpha_T \sqrt{2d \log(MT)}$.*

Theorem 2 establishes $\widetilde{\mathcal{O}}(d^{3/2}\sqrt{T})$ worst-case regret, which matches the regret bounds of TS methods for linear contextual bandits [5, 3] up to logarithmic factor. The regret bound shows no dependence on $N$, and has an additional $\mathcal{O}(\sqrt{\log \log K})$ dependence due to optimistic sampling which is very small for any reasonable assortment size $K$. Compared to Theorem 1, the additional factor $\sqrt{d}$ comes from the deviation of the random sampling which is addressed in the worst-case regret analysis.

The proof of Theorem 2 utilizes the anti-concentration property of the maximum of Gaussian random variables for ensuring frequent optimism. In particular, we show in the following lemma that the proposed optimistic sampling can ensure a constant probability of optimism.

**Lemma 2.** *Suppose $\|\hat{\theta}_t - \theta^*\|_{V_t} \leq \frac{1}{2\kappa}\sqrt{d \log\left(1 + \frac{tK}{d\lambda}\right) + 4\log t} + \frac{\sqrt{\lambda}}{\kappa}$ and we take optimistic samples of size $M = \lceil 1 - \frac{\log K}{\log(1 - 1/(4\sqrt{e\pi}))} \rceil$. Then we have*

$$\mathbb{P}\left( \widetilde{R}_t(S_t) > R_t(S_t^*, \theta^*) \mid \mathcal{F}_t \right) \geq \frac{1}{4\sqrt{e\pi}}.$$

The inverse of the lower-bounding probability $4\sqrt{e\pi}$ can be interpreted as the expected time between any two optimistic assortment selections. In other words, our modified algorithm is optimistic at least with a constant frequency. Then, using this frequent optimism, we can ensure that the cumulative regret due to the random sampling can be bounded. Along with this result, we show the concentrations of both regularized MLE and TS samples to establish the regret bound in Theorem 2. The proofs are left to Appendix D.

# 7 Numerical Study

In this section, we perform numerical evaluations to analyze two variants of our proposed algorithm: TS-MNL with optimistic sampling (Algorithm 2) and TS-MNL with the Gaussian approximation for the posterior distribution. We perform both synthetic experiments as well as simulated experiments using a real-world dataset: *MovieLens* dataset.[2] We simulated instances of the MNL contextual bandit problem with varying parameter values.

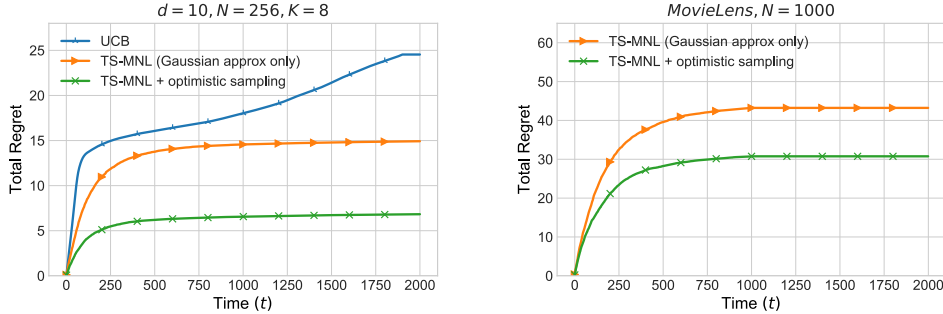

Figure 1: Regret growth with $T$ for a UCB method and TS-MNL variants on MNL contextual bandits.

We report the worst-case cumulative expected regret for each of the experiments. For the synthetic experiments, we randomly draw $\theta^*$ for each instance and hence we can directly compute the expected regret using $\theta^*$. For the experiments using MovieLens dataset, we use offline regression using the entire dataset to estimate the unknown parameter $\theta^*$ and compare with the estimates from online experiments. The details of the experimental setup and additional experimental results are presented in Appendix G.

Figure 1 shows the performances averaged over 40 independent instances for each experiment. For comparison, we evaluate the performances of our TS-MNL algorithms along with the performances of the UCB method proposed in [15]. The performances of the proposed two variants of TS-MNL are observed to be superior to that of the UCB method on the synthetic data in our experiments, which is consistent with the other empirical evidence of TS methods in the literature. The experiments with MovieLens dataset (and the additional experiments shown in Appendix G) suggest that our methods can be used and effective for problem instances with a large number of items, i.e., large $N$. Furthermore, TS-MNL with optimistic sampling consistently performs better than TS-MNL with Gaussian approximation only. The results of these experiments support our theoretical analysis: TS-MNL with optimistic sampling takes advantage of the MNL structure and can guarantee a worst-case statistical efficiency.

# 8 Discussions

In this paper, we study the dynamic assortment selection problem under an MNL model with contextual information. We propose two TS algorithms for the MNL contextual bandits which learn the parameters of the underlying choice model while simultaneously maximizing the cumulative revenue. We provide their theoretical performance bounds and show attractive numerical performances in our experiments. We also discuss the challenges which arise in worst-case regret analysis for this combinatorial action selection problem under the MNL model. We believe that these challenges are potentially present in many other problems involving combinatorial action selections with context/feature information beyond the MNL model. To our knowledge, the worst-case regret analysis in this work is the first frequentist regret guarantee for contextual bandits with combinatorial action selection of any kind. We believe that our proposed optimistic sampling framework can be useful for other combinatorial contextual bandit problems.

## Footnotes

[1] $\widetilde{\mathcal{O}}$ suppresses logarithmic dependence.

[2]https://grouplens.org/datasets/movielens/

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
