[Supplementary Material · NeurIPS_2019_TS_for_MNL_contextual_bandits-appendix_included.pdf]

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

# Appendices for Thompson Sampling for MNL Contextual Bandits

## A   Regularized Maximum Likelihood Estimation for MNL Model

We briefly discuss regularized maximum likelihood estimation (MLE) for MNL model – specifically the estimation of the unknown parameter $\theta^*$ of the MNL model with the rigde penalty. First, recall that $y_t \in \{0,1\}^{|S_t|}$ is the user choice where $y_{ti}$ is the $i$-th component of $y_t$. Then, the likelihood function under parameter $\theta$ is then given by

$$\mathcal{L}(\mathcal{D}_n|\theta) = \prod_{t=1}^{n} \prod_{i \in S_t} (p_{ti}(S_t, \theta))^{y_{ti}}$$

where $\mathcal{D}_n = \{X_t, S_t, y_t\}_{t=1}^{n}$ and $X_t = \{x_{ti}\}_{i \in [N]}$. Taking the negative logarithm gives

$$\ell_n(\theta) = -\log \mathcal{L}(\mathcal{D}_n|\theta) = -\sum_{t=1}^{n} \sum_{i \in S_t} y_{ti} \log p_{ti}(S_t, \theta)$$

which is known as the cross-entropy error function for the multi-class classification problem. Now, the ridge penalized maximum likelihood estimation for MNL model is given by the following minimization problem:

$$\hat{\theta} = \arg \min_{\theta} \left[ \ell_n(\theta) + \frac{\lambda}{2} \|\theta\|^2 \right] \tag{4}$$

with the penalty parameter $\lambda \geq 1$.

Taking the gradient of this penalized log-likelihood function with respect to $\theta$, we obtain

$$\nabla_\theta \left[ \ell_n(\theta) + \frac{\lambda}{2} \|\theta\|_2^2 \right] = \sum_{t=1}^{n} \sum_{i \in S_t} (p_{ti}(S_t, \theta) - y_{ti}) x_{ti} + \lambda\theta. \tag{5}$$

Instead of using the regularized MLE for the parameter estimation, one could consider using the MLE without regularization. For this, however, one may consider performing a random initialization (random exploration) to ensure that the matrix $V_t$ is invertible. This necessity comes from the classical likelihood theory [28]: as the sample size $n$ goes to infinity, we know the MLE $\hat{\theta}_n^{\text{ML}}$ is asymptotically normal, with $\hat{\theta}_n^{\text{ML}} - \theta^* \to \mathcal{N}(0, \mathcal{I}_{\theta^*}^{-1})$ where $\mathcal{I}_{\theta^*}$ is the Fisher information matrix. In the MNL model, $\mathcal{I}_{\theta^*}$ is lower bounded by $\sum_t \sum_{i \in S_t} p_{ti}(S_t, \theta^*) p_{t0}(S_t, \theta^*) x_{ti} x_{ti}^\top$ (see Lemma 4). Hence, if $p_{ti}(S_t, \theta^*) p_{t0}(S_t, \theta^*) \geq \kappa > 0$, then we can ensure that $\mathcal{I}_{\theta^*}$ is invertible and prevent asymptotic variance of $x^\top \hat{\theta}_n^{\text{ML}}$ from going to infinity for any $x$. When performing random exploration instead of the regularization, the length of such exploration needs to be specified to ensure that the minimum eigenvalue of the matrix $V_t$ is large enough — we discuss in detail in Appendix F.

## B   Proof of Theorem 1: Bayesian Regret Analysis

Let $\mathcal{F}_t$ denote the filtration which contains all available information up to round $t$. Recall that $\widetilde{\theta}_t$ is independently drawn from the posterior distribution $Q_t$ in Algorithm 1 and also note that in our Bayesian setting the posterior belief in $\theta^*$ is distributed as $Q_t$ conditioning on $\mathcal{F}_t$. Therefore, conditioning on $\mathcal{F}_t$, $\widetilde{\theta}_t$ and $\theta_t^*$ are i.i.d. with $Q_t$. Also note that our optimization oracle is a fixed combinatorial optimization algorithm and $\{x_{ti}\}_{i \in [N]}$ are fixed given $\mathcal{F}_t$. Hence, conditioning on $\mathcal{F}_t$, $S_t$ and $S^*$ are also i.i.d.

### B.1   Confidence Bound for Expected Revenue

We define a upper confidence expected revenue as

$$U_t(S, \hat{\theta}_t) = \frac{\sum_{i \in S} r_{ti} \exp\left\{ x_{ti}^\top \hat{\theta}_t + \alpha_t \|x_{ti}\|_{V_t^{-1}} \right\}}{1 + \sum_{j \in S} \exp\left\{ x_{tj}^\top \hat{\theta}_t + \alpha_t \|x_{tj}\|_{V_t^{-1}} \right\}}$$

where $\alpha_t > 0$ is the confidence width and its value is specified later (Lemma 4). Also, we define $V_t = \sum_{\tau=1}^t \sum_{i \in S_\tau} x_{\tau i} x_{\tau i}^\top$. Note that this upper confidence expected revenue $U_t$ is constructed for the sake of the analysis presented in this section and does not affect the proposed algorithm (or its assortment selection). We first decompose the immediate regret using $U_t$.

$$\mathbb{E}[\mathcal{R}(t) \mid \mathcal{F}_t] = \mathbb{E}\big[R_t(S_t^*, \theta^*) - R_t(S_t, \theta^*) \mid \mathcal{F}_t\big]$$
$$= \mathbb{E}\big[R_t(S_t^*, \theta^*) - U_t(S_t^*, \hat{\theta}_t) \mid \mathcal{F}_t\big] + \mathbb{E}\big[U_t(S_t^*, \hat{\theta}_t) - U_t(S_t, \hat{\theta}_t) \mid \mathcal{F}_t\big]$$
$$+ \mathbb{E}\big[U_t(S_t, \hat{\theta}_t) - R_t(S_t, \theta^*) \mid \mathcal{F}_t\big].$$

Notice that $\mathbb{E}\big[U_t(S_t^*, \hat{\theta}_t) - U_t(S_t, \hat{\theta}_t) \mid \mathcal{F}_t\big] = 0$ since conditioning on $\mathcal{F}_t$, $S_t$ and $S^*$ are i.i.d. and $U_t$ is a deterministic function. Hence, for the Bayesian cumulative regret, we are left bound the two quantities $\mathcal{R}^1_{\text{Bayes}}(T)$ and $\mathcal{R}^2_{\text{Bayes}}(T)$ as the following:

$$\sum_{t=1}^T \mathbb{E}[\mathcal{R}(t) \mid \mathcal{F}_t] = \underbrace{\sum_{t=1}^T \mathbb{E}\big[R_t(S_t^*, \theta^*) - U_t(S_t^*, \hat{\theta}_t) \mid \mathcal{F}_t\big]}_{\mathcal{R}^1_{\text{Bayes}}(T)} + \underbrace{\sum_{t=1}^T \mathbb{E}\big[U_t(S_t, \hat{\theta}_t) - R_t(S_t, \theta^*) \mid \mathcal{F}_t\big]}_{\mathcal{R}^2_{\text{Bayes}}(T)}$$

In the following sections, we present the upper-bounds for $\mathcal{R}^1_{\text{Bayes}}(T)$ and $\mathcal{R}^2_{\text{Bayes}}(T)$. Then we combine the results to establish the Bayesian cumulative regret for TS-MNL (Algorithm 1).

## B.2 Bounding $\mathcal{R}^1_{\text{Bayes}}(T)$

Before we present the upper bound for $\mathcal{R}^1_{\text{Bayes}}(T)$, we introduce the following lemma which utilizes the structure of the MNL model. Lemma 3 shows that the expected revenue $R_t$ (and hence $U_t$) has a Lipschitz property, i.e., Lemma 3 ensures that we can control the difference between expected revenues by bounding with maximum difference in utilities.

**Lemma 3.** *For any two utility parameters $u_t = [u_{t1}, ..., u_{tN}]$ and $u_t' = [u_{t1}', ..., u_{tN}']$, we have*

$$\frac{\sum_{i \in S} r_{ti} \exp(u_{ti})}{1 + \sum_{j \in S} \exp(u_{tj})} - \frac{\sum_{i \in S} r_{ti} \exp(u_{ti}')}{1 + \sum_{j \in S} \exp(u_{tj}')} \le \max_{i \in S} |u_{ti} - u_{ti}'|.$$

*In particular, if $u_{ti} \ge u_{ti}'$ for all $i$, then*

$$\frac{\sum_{i \in S} r_{ti} \exp(u_{ti})}{1 + \sum_{j \in S} \exp(u_{tj})} - \frac{\sum_{i \in S} r_{ti} \exp(u_{ti}')}{1 + \sum_{j \in S} \exp(u_{tj}')} \le \max_{i \in S} (u_{ti} - u_{ti}').$$

Note that in the statement of Lemma 3 we use the explicit form of expected revenues (with generic utility parameters) in order to accommodate both $R_t$ and $U_t$. Now, Lemma 4 below shows that the true parameter $\theta^*$ lies within an ellipsoid centered at $\hat{\theta}_t$ with confidence radius $\alpha_t$. This is the result for the non-i.i.d. finite-sample confidence bound for the MNL parameter.

**Lemma 4.** *Define $\alpha_t = \frac{1}{2\kappa} \sqrt{d \log\left(1 + \frac{tK}{d\lambda}\right) + 4 \log t} + \frac{\sqrt{\lambda}}{\kappa}$. If $\hat{\theta}_t$ is the solution to the regularized MLE in Eq.(4) at round $t$, then*

$$\|\hat{\theta}_t - \theta^*\|_{V_t} \le \alpha_t$$

*holds for all $t$ with a probability $1 - \mathcal{O}\left(\frac{1}{t^2}\right)$.*

If $\theta^*$ is indeed within the confidence region for all $t$, i.e., if the high probability event of Lemma 4 holds, then one can show that $x_{ti}^\top \hat{\theta}_t + \alpha_t \|x_{ti}\|_{V_t^{-1}} \ge x_{ti}^\top \theta^*$ for all $i$. Hence, $U_t(S_t^*, \hat{\theta}_t)$ is greater than $R_t(S_t^*, \theta^*)$. Then, $\mathcal{R}^1_{\text{Bayes}}(T)$ can be upper-bounded by 0. However, there is a small probability of failure for the confidence region which we need to take into consideration. The following lemma formally state the result.

**Lemma 5.** *Let the upper confidence expected revenue $U_t(S_t^*, \hat{\theta}_t)$ be defined with the confidence width $\alpha_t = \frac{1}{2\kappa} \sqrt{d \log\left(1 + \frac{tK}{d\lambda}\right) + 4 \log t} + \frac{\sqrt{\lambda}}{\kappa}$. Then, we have*

$$\sum_{t=1}^T \mathbb{E}\big[R_t(S_t^*, \theta^*) - U_t(S_t^*, \hat{\theta}_t) \mid \mathcal{F}_t\big] = \mathcal{O}(1).$$

## B.3 Bounding $\mathcal{R}^2_{\text{Bayes}}(T)$

This portion of the regret is controlled by the concentration of the upper-confidence expected revenue $U_t(S_t, \hat{\theta}_t)$ to the true expected revenue $R_t(S_t, \theta^*)$. We can first use Lemma 3 to upper-bound $\mathcal{R}^2_{\text{Bayes}}(T)$ by the expected maximum difference in utilities. Now, suppose that $\theta^*$ resides within the confidence region with the radius $\alpha_t$ for all rounds $t$ (Lemma 4). Then the same holds for the radius $\alpha_T$ since $\alpha_T \geq \alpha_t$. Using this fact and Cauchy-Schwartz inequality, we can further bound $\mathcal{R}^2_{\text{Bayes}}(T)$ by Eq.(6).

$$\sum_{t=1}^{T} \mathbb{E}\left[U_t(S_t, \hat{\theta}_t) - R_t(S_t, \theta^*) \mid \mathcal{F}_t\right] \leq \sum_{t=1}^{T} \mathbb{E}\left[\max_{i \in S_t}\left(x_{ti}^\top \hat{\theta}_t + \alpha_t \|x_{ti}\|_{V_t^{-1}} - x_{ti}^\top \theta^*\right) \mid \mathcal{F}_t\right]$$

$$\leq 2\alpha_T \sum_{t=1}^{T} \mathbb{E}\left[\max_{i \in S_t} \|x_{ti}\|_{V_t^{-1}} \mid \mathcal{F}_t\right] \qquad (6)$$

Then, we are left to control the sum of the expectations in Eq.(6). Specifically, we provide a worst-case bound on $\sum_{t=1}^{T} \max_{i \in S_t} \|x_{ti}\|_{V_t^{-1}}$ for any realization of random variables in Lemma 6, which presents a self-normalized bound.

**Lemma 6.** *Define $V_T = V + \sum_{t=1}^{T} \sum_{i \in S_t} x_{ti} x_{ti}^\top$ where $V = \lambda I_d$. Then we have*

$$\sum_{t=1}^{T} \max_{i \in S_t} \|x_{ti}\|_{V_t^{-1}} \leq \sqrt{2dT \log\left(1 + \frac{TK}{d\lambda}\right)}.$$

Combining the results of Lemma 6 and Eq.(6), we have

$$\sum_{t=1}^{T} \mathbb{E}\left[U_t(S_t, \hat{\theta}_t) - R_t(S_t, \theta^*) \mid \mathcal{F}_t\right] \leq 2\alpha_T \sqrt{2dT \log\left(1 + \frac{TK}{d\lambda}\right)} + \mathcal{O}(1)$$

where $\alpha_T = \frac{1}{2\kappa}\sqrt{d \log\left(1 + \frac{TK}{d\lambda}\right) + 4\log T} + \frac{\sqrt{\lambda}}{\kappa}$ and $\mathcal{O}(1)$ comes from the failure event of the concentration of $\hat{\theta}_t$ in Lemma 4.

## B.4 Combining $\mathcal{R}^1_{\text{Bayes}}(T)$ and $\mathcal{R}^2_{\text{Bayes}}(T)$

Combining the bounds for $\mathcal{R}^1_{\text{Bayes}}(T)$ and $\mathcal{R}^2_{\text{Bayes}}(T)$, we have

$$\mathcal{R}_{\text{Bayes}}(T) \leq \mathcal{O}(1) + \left[\frac{1}{\kappa}\sqrt{2d \log\left(1 + \frac{TK}{d\lambda}\right) + 2\log T} + \frac{\sqrt{\lambda}}{\kappa}\right] \cdot \sqrt{2dT \log\left(1 + \frac{TK}{d\lambda}\right)}.$$

For completeness, we choose $\lambda = d$ to get the regret bound shown in Theorem 1 which gives the Bayesian regret $\mathcal{R}_{\text{Bayes}}(T) = \mathcal{O}\left(d\sqrt{T} \log\left(1 + \frac{TK}{d^2}\right)\right)$. Since Algorithm 1 itself does not use the regularized MLE for parameter estimation, one may optimize over the choice of $\lambda$ in the regret bound.

# C  Proofs of Lemmas for Theorem 1

## C.1  Proof of Lemma 3

*Proof.* By the mean value theorem, there exists $\bar{u}_{ti} := (1-c)u_{ti} + cu'_{ti}$ for some $c \in (0, 1)$ with

$$\frac{\sum_{i \in S} r_{ti} \exp(u_{ti})}{1 + \sum_{j \in S} \exp(u_{tj})} - \frac{\sum_{i \in S} r_{ti} \exp(u'_{ti})}{1 + \sum_{j \in S} \exp(u'_{tj})}$$

$$= \sum_{i \in S} r_{ti} p_{ti}(S, \bar{u}_t)(u_{ti} - u'_{ti}) - R_t(S, \bar{u}_t) \cdot \sum_{i \in S} p_{ti}(S, \bar{u}_t)(u_{ti} - u'_{ti})$$

$$= \sum_{i \in S} \left(r_{ti} - R_t(S, \bar{u}_t)\right) p_{ti}(S, \bar{u}_t)(u_{ti} - u'_{ti})$$

$$\leq \max_{i \in S} |u_{ti} - u'_{ti}|$$

where the inequality is from $|r_{ti}| \leq 1$, and $p_{ti}(S, \bar{u}_t) \leq 1$ is a multinomial probability (and hence $R_t(S, \bar{u}_t) \leq 1$).  $\square$

## C.2  Proof of Lemma 4

*Proof.* We first define the function $G_n(\theta)$ which we use throughout the proof:

$$G_n(\theta) = \sum_{t=1}^{n} \sum_{i \in S_t} [(p_{ti}(S_t, \theta) - p_{ti}(S_t, \theta^*)) x_{ti}] + \lambda(\theta - \theta^*)$$

$G_n(\theta)$ is the difference in the gradients of the ridge penalized maximum likelihood in Eq.(5) evaluated at $\theta$ and at $\theta^*$. Notice that $G_n(\hat{\theta}) = \sum_{t=1}^{n} \sum_{i \in S_t} \epsilon_{ti} x_{ti} - \lambda \theta^*$ since the choice of $\hat{\theta}$ is given by the ridge penalized maximum likelihood. To see that, first note that $\hat{\theta}$ is the minimizer of Eq.(4); hence is given by the solution to the following equation:

$$\sum_{t=1}^{n} \sum_{i \in S_t} \left( p_{ti}(S_t, \hat{\theta}) - y_{ti} \right) x_{ti} + \lambda \hat{\theta} = 0 \tag{7}$$

Therefore, it follows that

$$G_n(\hat{\theta}) = \sum_{t=1}^{n} \sum_{i \in S_t} \left( p_{ti}(S_t, \hat{\theta}) - p_{ti}(S_t, \theta^*) \right) x_{ti} + \lambda(\hat{\theta} - \theta^*)$$

$$= \sum_{t=1}^{n} \sum_{i \in S_t} \left( p_{ti}(S_t, \hat{\theta}) - y_{ti} \right) x_{ti} + \lambda \hat{\theta} + \sum_{t=1}^{n} \sum_{i \in S_t} (y_{ti} - p_{ti}(S_t, \theta^*)) x_{ti} - \lambda \theta^*$$

$$= 0 + \sum_{t=1}^{n} \sum_{i \in S_t} \epsilon_{ti} x_{ti} - \lambda \theta^*$$

where the last equality is from (7) and the definition of $\epsilon_{ti} = y_{ti} - p_{ti}(S_t, \theta^*)$. For convenience, we define $Z_n := \sum_{t=1}^{n} \sum_{i \in S_t} \epsilon_{ti} x_{ti}$. Hence, $G_n(\hat{\theta}) = Z_n - \lambda \theta^*$. Also, we will denote $p_{ti}(\theta) := p_{ti}(S_t, \theta)$ when it is clear that $S_t$ is the assortment chosen at round $t$.

For any $\theta_1, \theta_2 \in \mathbb{R}^d$, the mean value theorem implies that there exists $\bar{\theta} = c\theta_1 + (1 - c)\theta_2$ with some $c \in (0, 1)$ such that

$$G_n(\theta_1) - G_n(\theta_2) = \sum_{t=1}^{n} \sum_{i \in S_t} \left[ (p_{ti}(\theta_1) - p_{ti}(\theta_2)) x_{ti} \right] + \lambda(\theta_1 - \theta_2)$$

$$= \left[ \left( \sum_{t=1}^{n} \sum_{i \in S_t} \sum_{j \in S_t} \nabla_j p_{ti}(\bar{\theta}) x_{ti} x_{tj}^\top \right) + \lambda I_d \right] (\theta_1 - \theta_2)$$

$$= \left[ \sum_{t=1}^{n} \left( \sum_{i \in S_t} p_{ti}(\bar{\theta}) x_{ti} x_{ti}^\top - \sum_{i \in S_t} \sum_{j \in S_t} p_{ti}(\bar{\theta}) p_{tj}(\bar{\theta}) x_{ti} x_{tj}^\top \right) + \lambda I_d \right] (\theta_1 - \theta_2)$$

where $I_d$ is a $d \times d$ identitiy matrix. We define the matrix $H_t$ as

$$H_t := \sum_{i \in S_t} p_{ti}(\bar{\theta}) x_{ti} x_{ti}^\top - \sum_{i,j \in S_t} p_{ti}(\bar{\theta}) p_{tj}(\bar{\theta}) x_{ti} x_{tj}^\top$$

Notice $H_t$ is a Hessian of a negative log-likelihood which is convex. Hence, $H_t$ is positive semidefinite. Also note that

$$(x_i - x_j)(x_i - x_j)^\top = x_i x_i^\top + x_j x_j^\top - x_i x_j^\top - x_j x_i^\top \succeq 0$$

which implies $x_i x_i^\top + x_j x_j^\top \succeq x_i x_j^\top + x_j x_i^\top$. Therefore, it follows that

$$
\begin{aligned}
H_t &= \sum_{i \in S_t} p_{ti}(\bar{\theta}) x_{ti} x_{ti}^\top - \sum_{i \in S_t} \sum_{j \in S_t} p_{ti}(\bar{\theta}) p_{tj}(\bar{\theta}) x_{ti} x_{tj}^\top \\
&= \sum_{i \in S_t} p_{ti}(\bar{\theta}) x_{ti} x_{ti}^\top - \frac{1}{2} \sum_{i \in S_t} \sum_{j \in S_t} p_{ti}(\bar{\theta}) p_{tj}(\bar{\theta}) \left( x_{ti} x_{tj}^\top + x_{tj} x_{ti}^\top \right) \\
&\succeq \sum_{i \in S_t} p_{ti}(\bar{\theta}) x_{ti} x_{ti}^\top - \frac{1}{2} \sum_{i \in S_t} \sum_{j \in S_t} p_{ti}(\bar{\theta}) p_{tj}(\bar{\theta}) \left( x_{ti} x_{ti}^\top + x_{tj} x_{tj}^\top \right) \\
&= \sum_{i \in S_t} p_{ti}(\bar{\theta}) x_{ti} x_{ti}^\top - \sum_{i \in S_t} \sum_{j \in S_t} p_{ti}(\bar{\theta}) p_{tj}(\bar{\theta}) x_{ti} x_{ti}^\top \\
&= \sum_{i \in S_t} p_{ti}(\bar{\theta}) \left( 1 - \sum_{j \in S_t} p_{tj}(\bar{\theta}) \right) x_{ti} x_{ti}^\top \\
&= \sum_{i \in S_t} p_{ti}(\bar{\theta}) p_{t0}(\bar{\theta}) x_{ti} x_{ti}^\top
\end{aligned}
$$

where $p_{t0}(\bar{\theta})$ is the probability of choosing the outside option. Now,

$$
\begin{aligned}
G_n(\theta_1) - G_n(\theta_2) &= \left[ \sum_{t=1}^n H_t + \lambda I_d \right] (\theta_1 - \theta_2) \\
&\geq \left[ \sum_{t=1}^n \sum_{i \in S_t} p_{ti}(\bar{\theta}) p_{t0}(\bar{\theta}) x_{ti} x_{ti}^\top + \lambda I_d \right] (\theta_1 - \theta_2) \\
&:= \mathcal{H}(\bar{\theta})(\theta_1 - \theta_2).
\end{aligned}
$$

Consider some $\bar{\theta} \in \mathbb{R}^d$. From Assumption 2, $p_{ti}(\bar{\theta}) p_{t0}(\bar{\theta})$ is lower-bounded by $\kappa$. Then we have

$$
(\theta_1 - \theta_2)^\top (G_n(\theta_1) - G_n(\theta_2)) \geq (\theta_1 - \theta_2)^\top (\kappa V_n)(\theta_1 - \theta_2) > 0
$$

for any $\theta_1 \neq \theta_2$. Therefore, $G_n(\theta)$ is an injection from $\mathbb{R}^d$ to $\mathbb{R}^d$, and so $G^{-1}$ is a well-defined function. By the definition, $G_n(\theta^*) = 0$. Hence, for any $\theta \in \mathbb{R}^d$, we have

$$
\begin{aligned}
\|G_n(\theta)\|_{V_n^{-1}}^2 &= \|G_n(\theta) - G_n(\theta^*)\|_{V_n^{-1}}^2 \\
&= (G_n(\theta) - G_n(\theta^*))^\top V_n^{-1} (G_n(\theta) - G_n(\theta^*)) \\
&\geq (\theta - \theta^*)^\top \mathcal{H}(\bar{\theta}) V_n^{-1} \mathcal{H}(\bar{\theta}) (\theta - \theta^*) \\
&\geq \kappa^2 (\theta - \theta^*)^\top V_n (\theta - \theta^*) \\
&= \kappa^2 \|\hat{\theta} - \theta^*\|_{V_n}^2
\end{aligned}
$$

where the last inequality is from $\mathcal{H}(\bar{\theta}) \succeq \kappa V_n$. Now, recall for $\hat{\theta}$ which is the solution to Eq.(7), $G_n(\hat{\theta}) = Z_n - \lambda \theta^*$ where $Z_n = \sum_{t=1}^n \sum_{i \in S_t} \epsilon_{ti} x_{ti}$. Hence, we have

$$
\kappa \|\hat{\theta} - \theta^*\|_{V_n} \leq \|G_n(\hat{\theta})\|_{V_n^{-1}} \leq \|Z_n\|_{V_n^{-1}} + \lambda \|\theta^*\|_{V_n^{-1}}
$$

Then we can use Theorem 1 in [1], which states if the noise $\epsilon_{ti}$ is sub-Gaussian with parameter $\sigma$ (with $\sigma = \frac{1}{2}$ in our problem), then

$$
\|Z_n\|_{V_n^{-1}}^2 \leq 2\sigma^2 \log \left( \frac{\det(V_n)^{1/2} \det(V)^{-1/2}}{\delta} \right)
$$

with probability at least $1 - \delta$. Then we combine with Lemma 9. So it follows that

$$
\|Z_n\|_{V_n^{-1}}^2 \leq 2\sigma^2 \left[ \frac{d}{2} \log \left( \frac{\text{trace}(V) + nK}{d} \right) - \frac{1}{2} \log \det(V) + \log \frac{1}{\delta} \right].
$$

Since $V = \lambda I_d$, it follows that

$$\|Z_n\|_{V_n^{-1}}^2 \le 2\sigma^2 \left[ \frac{d}{2} \log\left( \frac{d\lambda + nK}{d} \right) - \frac{1}{2} \log \lambda^d + \log \frac{1}{\delta} \right]$$

$$= 2\sigma^2 \left[ \frac{d}{2} \log\left( \lambda + \frac{nK}{d} \right) - \frac{d}{2} \log \lambda + \log \frac{1}{\delta} \right]$$

$$= 2\sigma^2 \left[ \frac{d}{2} \log\left( 1 + \frac{nK}{d\lambda} \right) + \log \frac{1}{\delta} \right].$$

Then for $\|\theta^*\|_{V_n^{-1}}$, we have

$$\|\theta^*\|_{V_n^{-1}}^2 \le \frac{\|\theta^*\|^2}{\lambda_{\min}(V_n)} \le \frac{\|\theta^*\|^2}{\lambda_{\min}(V)} \le \frac{\|\theta^*\|^2}{\lambda}.$$

Hence, $\lambda \|\theta^*\|_{V_n^{-1}} \le \sqrt{\lambda}$ since $\|\theta^*\| \le 1$. Combining the results and using the fact that $\sigma = \frac{1}{2}$ for our problem, we have that

$$\|\hat{\theta}_n - \theta^*\|_{V_n} \le \frac{1}{2\kappa} \sqrt{d \log\left( 1 + \frac{nK}{d\lambda} \right) + 2 \log \frac{1}{\delta}} + \frac{\sqrt{\lambda}}{\kappa}.$$

with probability at least $1 - \delta$.

$\square$

### C.3 Proof of Lemma 5

*Proof.* First, define event $\hat{\mathcal{E}}_t = \{ \|\theta^* - \hat{\theta}_t\|_{V_t} \le \alpha_t \}$, i.e. the regularized MLE estimate concentrates properly to $\theta^*$ in rounds $t$. From Lemma 4, this concentration event holds with probability $1 - \mathcal{O}\left( \frac{1}{t^2} \right)$ for each round $t$. On $\hat{\mathcal{E}}_t$, we show $x_{ti}^\top \theta^* \le x_{ti}^\top \hat{\theta}_t + \alpha_t \|x_{ti}\|_{V_t^{-1}}$ for all $i$.

$$|x_{ti}^\top \hat{\theta}_t - x_{ti}^\top \theta^*| = \left| \left[ V_t^{-1/2}(\hat{\theta}_t - \theta^*) \right]^\top (V_t^{-1/2} x_{ti}) \right|$$

$$\le \left\| V_t^{-1/2}(\hat{\theta}_t - \theta^*) \right\| \left\| V_t^{-1/2} x_{ti} \right\|$$

$$= \|\hat{\theta}_t - \theta^*\|_{V_t} \|x_{ti}\|_{V_t^{-1}}$$

$$\le \alpha_t \|x_{ti}\|_{V_t^{-1}}$$

where the first inequality is by Hölder's inequality. Hence, it follows that

$$x_{ti}^\top \theta^* - \left( x_{ti}^\top \hat{\theta}_t + \alpha_t \|x_{ti}\|_{V_t^{-1}} \right) \le 0$$

for all $i$. Hence, using the restricted monotonicity in Lemma 1, if event $\hat{\mathcal{E}}_t$ holds, then we have

$$R_t(S_t^*, \theta^*) - U_t(S_t^*, \hat{\theta}_t) \le 0.$$

Then we have

$$\mathbb{E}\left[ R_t(S_t^*, \theta^*) - U_t(S_t^*, \hat{\theta}_t) \mid \mathcal{F}_t \right] \le \mathbb{E}\left[ \left( R_t(S_t^*, \theta^*) - U_t(S_t^*, \hat{\theta}_t) \right) \mathbb{1}(\hat{\mathcal{E}}_t) \mid \mathcal{F}_t \right] + \mathbb{E}\left[ \mathbb{1}(\hat{\mathcal{E}}_t^c) \mid \mathcal{F}_t \right]$$

$$\le 0 + \mathcal{O}(t^{-2}).$$

Therefore, summing over all $t \le T$, we have

$$\sum_{t=1}^T \mathbb{E}\left[ R_t(S_t^*, \theta^*) - U_t(S_t^*, \hat{\theta}_t) \mid \mathcal{F}_t \right] \le 0 + \sum_{t=1}^T \mathcal{O}(t^{-2}) = \mathcal{O}(1).$$

$\square$

## C.4 Proof of Lemma 6

The proof of Lemma 6 requires the following three technical lemmas.

**Lemma 7.** *Let $x_{ti} \in \mathbb{R}^d$. Then we have*

$$\det\left(I + \sum_{i=1}^{n} x_{ti}x_{ti}^{\top}\right) \geq 1 + \sum_{i=1}^{n} \|x_{ti}\|_2^2 \tag{8}$$

*Proof.* Let $\lambda_1, \lambda_2, ..., \lambda_d$ be the eigenvalues of $\sum_{i=1}^{n} x_{ti}x_{ti}^{\top}$. Since $\sum_{i=1}^{n} x_{ti}x_{ti}^{\top}$ is positive semi-definite, $\lambda_j \geq 0$ for all $j$. Hence,

$$\det\left(I + \sum_{i=1}^{n} x_{ti}x_{ti}^{\top}\right) = \prod_{j=1}^{d}(1 + \lambda_j)$$

$$\geq 1 + \sum_{j=1}^{d} \lambda_j$$

$$= 1 - d + \sum_{j=1}^{d}(1 + \lambda_j)$$

$$= 1 - d + \text{trace}\left(I + \sum_{i=1}^{n} x_{ti}x_{ti}^{\top}\right)$$

$$= 1 - d + d + \sum_{i=1}^{n} \|x_{ti}\|_2^2$$

$\square$

**Lemma 8.** *Suppose $\|x_{ti}\| \leq 1$ for all $t$ and $i$. Define $V_t = V + \sum_{\tau=1}^{t} \sum_{i \in S_\tau} x_{\tau i}x_{\tau i}^{\top}$ with $V = \lambda I_d$. If $\lambda \geq 1$, then*

$$\sum_{\tau=1}^{t} \max_{i \in S_\tau} \|x_{\tau i}\|_{V_\tau^{-1}}^2 \leq 2\log\left(\frac{\det(V_t)}{\lambda_{\min}(V)^d}\right).$$

*Proof.*

$$\det(V_t) = \det\left(V_{t-1} + \sum_{i \in S_t} x_{ti}x_{ti}^{\top}\right)$$

$$= \det(V_{t-1})\det\left(I + \sum_{i \in S_t} V_{t-1}^{-1/2}x_{ti}(V_{t-1}^{-1/2}x_{ti})^{\top}\right)$$

$$\geq \det(V_{t-1})\left(1 + \sum_{i \in S_t} \|x_{ti}\|_{V_{t-1}^{-1}}^2\right)$$

$$\geq \det(V)\prod_{\tau=1}^{t}\left(1 + \sum_{i \in S_\tau} \|x_{\tau i}\|_{V_\tau^{-1}}^2\right)$$

$$\geq \det(V)\prod_{\tau=1}^{t}\left(1 + \max_{i \in S_\tau} \|x_{\tau i}\|_{V_\tau^{-1}}^2\right) \tag{9}$$

The first inequality comes from Lemma 7. The second inequality comes from applying the first inequality repeatedly.

Let $\lambda_{\min}(V_t)$ be the minimum eigenvalue of $V_t$. We have

$$\max_{i \in S_t} \|x_{ti}\|_{V_t^{-1}}^2 \leq \max_{i \in S_t} \frac{\|x_{ti}\|^2}{\lambda_{\min}(V_t)} \leq \frac{1}{\lambda_{\min}(V)} = \frac{1}{\lambda}.$$

Since $\lambda \geq 1$, using the fact that $z \leq 2\log(1+z)$ for any $z \in [0,1]$, we have

$$\sum_{\tau=1}^{t} \max_{i \in S_\tau} \|x_{\tau i}\|_{V_\tau^{-1}}^2 \leq 2\sum_{\tau=1}^{t} \log\left(1 + \max_{i \in S_\tau} \|x_{\tau i}\|_{V_\tau^{-1}}^2\right)$$

$$= 2\log\prod_{\tau=1}^{t}\left(1 + \max_{i \in S_\tau} \|x_{\tau i}\|_{V_\tau^{-1}}^2\right)$$

$$\leq 2\log\left(\frac{\det(V_t)}{\det(V)}\right)$$

The last inequality is from (9)

$\square$

**Lemma 9.** *Suppose* $\|x_{ti}\| \leq 1$ *for all t. Then* $\det(V_t)$ *is increasing with respect to t and*

$$\det(V_t) \leq \left(\frac{trace(V) + tK}{d}\right)^d \tag{10}$$

*Proof.* For any symmetric positive definite matrix $\widetilde{V} \in \mathbb{R}^{d \times d}$ and column vector $x \in \mathbb{R}^d$, we have

$$\det(\widetilde{V} + xx^\top) = \det(V)\det\left(I + \widetilde{V}^{-1/2}xx^\top\widetilde{V}^{-1/2}\right)$$

$$= \det(\widetilde{V})\det(1 + \|\widetilde{V}^{-1/2}x\|^2)$$

$$\geq \det(\widetilde{V}).$$

The second equality above is due to Sylvester's determinant theorem, which states that $\det(I+BA) = \det(I + AB)$. Let $\lambda_1, ..., \lambda_d > 0$ be the eigenvalues of $V_t$. Then

$$\det(V_t) \leq \left(\frac{\lambda_1 + ... + \lambda_d}{d}\right)^d$$

$$= \left(\frac{\text{trace}(V_t)}{d}\right)^d$$

$$= \left(\frac{\text{trace}(V) + \sum_{\tau=1}^{t}\sum_{i \in S_\tau}\text{trace}(x_{\tau i}x_{ti}^\top)}{d}\right)^d$$

$$= \left(\frac{\text{trace}(V) + \sum_{\tau=1}^{t}\sum_{i \in S_\tau}\|x_{\tau i}\|^2}{d}\right)^d$$

$$\leq \left(\frac{\text{trace}(V) + tK}{d}\right)^d.$$

$\square$

*Proof of Lemma 6.* Combining Lemma 8 and Lemma 9, we have that

$$\sum_{\tau=1}^{t}\max_{i \in S_\tau}\|x_{\tau i}\|_{V_\tau^{-1}}^2 \leq 2\log\left(\frac{\det(V_t)}{\det(V)}\right)$$

$$\leq 2\log\left[\left(\frac{\text{trace}(V) + tK}{d}\right)^d \frac{1}{\det(V)}\right]$$

$$\leq 2d\log\left(1 + \frac{tK}{d\lambda}\right).$$

Then applying the Cauchy-Schwarz inequality, we have

$$\sum_{\tau=1}^{t}\max_{i \in S_\tau}\|x_{\tau i}\|_{V_\tau^{-1}} = \sqrt{2dt\log\left(1 + \frac{tK}{d\lambda}\right)}.$$

$\square$

# D Proof of Theorem 2: Worst-case Regret Analysis

We first decompose the cumulative regret, similar to the procedure in previous sections but this time using $\widetilde{R}_t(S_t)$. In the following sections, we derive the bounds for $\mathcal{R}_1(t)$ and $\mathcal{R}_2(t)$ separately.

$$\mathcal{R}(T) = \underbrace{\sum_{t=1}^{T} \mathbb{E}[R_t(S_t^*, \theta^*) - \widetilde{R}_t(S_t)]}_{\mathcal{R}_1(T)} + \underbrace{\sum_{t=1}^{T} \mathbb{E}[\widetilde{R}_t(S_t) - R_t(S_t, \theta^*)]}_{\mathcal{R}_2(T)}$$

## D.1 Bounding $\mathcal{R}_2(T)$.

We can control $\mathcal{R}_2(T)$ by showing that both MLE $\hat{\theta}_t$ and TS parameters $\{\widetilde{\theta}_t\}$ concentrate appropriately. To show each of these concentration results, we first further decompose $\mathcal{R}_2(T)$:

$$\mathcal{R}_2(T) = \sum_{t=1}^{T} \mathbb{E}[\widetilde{R}_t(S_t) - R_t(S_t, \hat{\theta}_t)] + \sum_{t=1}^{T} \mathbb{E}[R_t(S_t, \hat{\theta}_t) - R_t(S_t, \theta^*)]. \tag{11}$$

The second term deals with the estimation error and can be bounded by the concentration of $\hat{\theta}_t$ in Lemma 4 and the Lipschitz-like property in Lemma 3, i.e., with probability $1 - \mathcal{O}(t^{-2})$, we have

$$R_t(S_t, \hat{\theta}_t) - R_t(S_t, \theta^*) \le \max_{i \in S_t} \left| x_{ti}^\top (\hat{\theta}_t - \theta^*) \right| \le \alpha_t \max_{i \in S_t} \|x_{ti}\|_{V_t^{-1}}. \tag{12}$$

The first term in Eq.(11) deals with the random sampling of $\{\widetilde{\theta}_t^{(j)}\}$. Again, we can bound the difference in expected revenue by the difference in utility estimates using Lemma 3: $\widetilde{R}_t(S_t) - R_t(S_t, \hat{\theta}_t) \le \max_{i \in S_t} (\widetilde{u}_{ti} - x_{ti}^\top \hat{\theta}_t)$. Then we are left to show that $\widetilde{u}_{ti}$ concentrates appropriately for all $i \in [N]$. The following lemma ensures the concentration of $\widetilde{u}_{ti}$.

**Lemma 10.** *Let $\beta_t = \alpha_t \min\left( \sqrt{4d \log(Mt)}, \sqrt{2 \log(2M)} + \sqrt{4 \log(Nt)} \right)$. Then for all $i \in [N]$,*

$$\widetilde{u}_{ti} - x_{ti}^\top \hat{\theta}_t \le \beta_t \|x_{ti}\|_{V_t^{-1}}.$$

*with probability $1 - \mathcal{O}\left(\frac{1}{t^2}\right)$.*

**Remark 1.** *Lemma 10 shows that the confidence radius $\beta_t$ is larger than $\alpha_t$ by the factor of at most $\sqrt{2d \log(Mt)}$. The additional $\sqrt{d}$ factor comes from the oversampling of TS, which also appears in other TS methods for linear contextual bandit problems [5, 3]. $\sqrt{\log M}$ factor comes from drawing optimistic samples where $M = \mathcal{O}(\log K)$; hence the marginal increase of the regret bound due to optimistic sampling is very small.*

Hence for the first term in Eq.(11), we have $\widetilde{R}_t(S_t) - R_t(S_t, \hat{\theta}_t) \le \beta_t \max_{i \in S_t} \|x_{ti}\|_{V_t^{-1}}$ with probability $1 - \mathcal{O}\left(\frac{1}{t^2}\right)$. We combine with Eq.(12) to derive the bound for $\mathcal{R}_2(T)$:

$$\mathcal{R}_2(T) \le \sum_{t=1}^{T} (\alpha_t + \beta_t) \max_{i \in S_t} \|x_{ti}\|_{V_t^{-1}} + \sum_{t=1}^{T} \mathcal{O}\left(t^{-2}\right) \tag{13}$$

## D.2 Bounding $\mathcal{R}_1(T)$.

As discussed in Section 5, a sufficient condition for ensuring the success of TS is to show the probability of TS samples being optimistic is high enough. The following lemma lower-bounds the probability that the expected revenue under sampled parameters is higher than the optimal expected revenue under the true parameter. The proof utilizes the anti-concentration property of Gaussian distribution.

**Lemma 2** (restate). *Suppose $\|\hat{\theta}_t - \theta^*\|_{V_t} \le \alpha_t$ and we take $M = \left\lceil 1 - \frac{\log K}{\log\left(1 - 1/(4\sqrt{e\pi})\right)} \right\rceil$ samples. Then we have*

$$\mathbb{P}\left( \widetilde{R}_t(S_t) > R_t(S_t^*, \theta_t^*) \mid \mathcal{F}_t \right) \ge \frac{1}{4\sqrt{e\pi}}. \tag{14}$$

Using this frequent optimistic sampling, we can ensure that the regret due to the oversampling is not too large.

**Lemma 12.** *Let* $\widetilde{p} = \frac{1}{4\sqrt{e\pi}}$*. Then, we have*

$$\sum_{t=1}^{T} \mathbb{E}[R_t(S_t^*, \theta_t^*) - \widetilde{R}_t(S_t)] \leq \frac{4\beta_T}{\widetilde{p}} \left( \sqrt{2dT \log\left(1 + \frac{TK}{d\lambda}\right)} + \sqrt{\frac{8T}{\lambda} \log 2T} \right) + \mathcal{O}(1)$$

### D.3 Combining the results

Applying Lemma 6 to the bound for $\mathcal{R}_2(T)$ in Eq.(13) and combining with Lemma 12, we have the final bound for the worst-case cumulative regret.

$$\mathcal{R}(T) \leq (\alpha_T + \beta_T)\sqrt{2dT \log(T/d)} + 16\sqrt{e\pi}\beta_T \left( \sqrt{2dT \log\left(1 + \frac{TK}{d\lambda}\right)} + \sqrt{\frac{8T}{\lambda} \log 2T} \right) + \mathcal{O}(1)$$

## E Proofs of Lemmas for Theorem 2

### E.1 Proof of Lemma 10

*Proof.* Given $\mathcal{F}_t$, each of Gaussian random variable $x_{ti}^\top \widetilde{\theta}_t^{(j)}$ has mean $x_{ti}^\top \hat{\theta}_t$ and standard deviation $\alpha_t \|x_{ti}\|_{V_t^{-1}}$.

$$|\widetilde{u}_{ti} - x_{ti}^\top \hat{\theta}_t| = \alpha_t \|x_{ti}\|_{V_t^{-1}} \frac{\left| \max_j x_{ti}^\top \widetilde{\theta}_t^{(j)} - x_{ti}^\top \hat{\theta}_t \right|}{\alpha_t \|x_{ti}\|_{V_t^{-1}}}$$

$$\leq \alpha_t \|x_{ti}\|_{V_t^{-1}} \max_j \left| \frac{x_{ti}^\top \widetilde{\theta}_t^{(j)} - x_{ti}^\top \hat{\theta}_t}{\alpha_t \|x_{ti}\|_{V_t^{-1}}} \right|$$

$$= \alpha_t \|x_{ti}\|_{V_t^{-1}} \max_j |Z_j|$$

where each $Z_j$ is a standard normal random variable. Using the result from Lemma 13, we have $\max_j |Z_j| \leq \sqrt{2\log(2M)} + \sqrt{4\log t}$ with probability at least $1 - \frac{1}{t^2}$. Then, for all $i \in [N]$,

$$|\widetilde{u}_{ti} - x_{ti}^\top \hat{\theta}_t| \leq \left( \sqrt{2\log(2M)} + \sqrt{4\log(Nt)} \right) \alpha_t \|x_{ti}\|_{V_t^{-1}}$$

with probability at least $1 - \frac{1}{t^2}$. Alternatively, let $m = \arg\max_j x_{ti}^\top \widetilde{\theta}_t^{(j)}$. Then we can write

$$|\widetilde{u}_{ti} - x_{ti}^\top \hat{\theta}_t| = \left| \max_j x_{ti}^\top \widetilde{\theta}_t^{(j)} - x_{ti}^\top \hat{\theta}_t \right|$$

$$= \left| x_{ti}^\top (\widetilde{\theta}_t^{(m)} - \hat{\theta}_t) \right|$$

$$= \left| x_{ti}^\top V_t^{-1/2} V_t^{1/2} (\widetilde{\theta}_t^{(m)} - \hat{\theta}_t) \right|$$

$$\leq \alpha_t \|x_{ti}\|_{V_t^{-1}} \left\| \alpha_t^{-1} V_t^{1/2} (\widetilde{\theta}_t^{(m)} - \hat{\theta}_t) \right\|$$

$$\leq \alpha_t \|x_{ti}\|_{V_t^{-1}} \max_j \left\| \alpha_t^{-1} V_t^{1/2} (\widetilde{\theta}_t^{(j)} - \hat{\theta}_t) \right\|$$

$$= \alpha_t \|x_{ti}\|_{V_t^{-1}} \max_j \|\zeta_j\|$$

where each element in $\zeta_j \in \mathbb{R}^d$ is a univariate standard normal variable $\mathcal{N}(0, 1)$. Hence, each $\|\zeta_j\| \leq \sqrt{4d\log t}$ with probability at least $1 - \frac{1}{t^2}$. Using the union bound for all $j \in \{1, ..., M\}$, we have with probability at least $1 - \frac{1}{t^2}$

$$|\widetilde{u}_{ti} - x_{ti}^\top \hat{\theta}_t| \leq \sqrt{4d\log(Mt)}\alpha_t \|x_{ti}\|_{V_t^{-1}}.$$

$\square$

**Lemma 13.** *Let $Z_i \sim \mathcal{N}(0,1), i = 1, ..., n$ be a standard Gaussian random variable. Then we have*

$$\mathbb{P}\left(\max_i |Z_i| \leq \sqrt{2\log(2n)} + \sqrt{2\log\frac{1}{\delta}}\right) \geq 1 - \delta.$$

*Proof.* Using the Chernoff bound, for each $Z_i$, we have

$$\mathbb{P}(|Z_i| > \epsilon) \leq 2e^{-\epsilon^2/2}.$$

Applying the union bound, we have

$$
\begin{aligned}
\mathbb{P}\left(\max_i |Z_i| > \sqrt{2\log(2n)} + \epsilon\right) &\leq 2n \exp\left(-(\sqrt{2\log(2n)} + \epsilon)^2/2\right) \\
&= 2n \exp(-\log(2n) - \epsilon\sqrt{2\log(2n)} - \epsilon^2/2) \\
&\leq e^{-\epsilon\sqrt{2\log(2n)}} e^{-\epsilon^2/2} \\
&\leq e^{-\epsilon^2/2}.
\end{aligned}
$$

Letting $\delta = e^{-\epsilon^2/2}$, we have the result. $\qquad\square$

### E.2 Proof of Lemma 2

*Proof.* Given $\mathcal{F}_t$, each of Gaussian random variable $x_{ti}^\top \widetilde{\theta}_t^{(j)}$ has mean $x_{ti}^\top \hat{\theta}_t$ and standard deviation $\alpha_t \|x_{ti}\|_{V_t^{-1}}$. Hence, for each $i \in S_t^*$, we have

$$
\begin{aligned}
\mathbb{P}\left(\max_j x_{ti}^\top \widetilde{\theta}_t^{(j)} > x_{ti}^\top \theta^* \mid \mathcal{F}_t\right) &= 1 - \mathbb{P}\left(x_{ti}^\top \widetilde{\theta}_t^{(j)} \leq x_{ti}^\top \theta^*, \forall j \in \{1, ..., M\} \mid \mathcal{F}_t\right) \\
&= 1 - \mathbb{P}\left(\frac{x_{ti}^\top \widetilde{\theta}_t^{(j)} - x_{ti}^\top \hat{\theta}_t}{\alpha_t \|x_{ti}\|_{V_t^{-1}}} \leq \frac{x_{ti}^\top \theta^* - x_{ti}^\top \hat{\theta}_t}{\alpha_t \|x_{ti}\|_{V_t^{-1}}}, \forall j \in \{1, ..., M\} \mid \mathcal{F}_t\right) \\
&= 1 - \mathbb{P}\left(Z_j \leq \frac{x_{ti}^\top \theta^* - x_{ti}^\top \hat{\theta}_t}{\alpha_t \|x_{ti}\|_{V_t^{-1}}}, \forall j \in \{1, ..., M\} \mid \mathcal{F}_t\right)
\end{aligned}
$$

where $Z_j$ is a standard normal random variable. By the assumption, we have $|x_{ti}^\top \theta^* - x_{ti}^\top \hat{\theta}_t| \leq \alpha_t \|x_{ti}\|_{V_t^{-1}}$ for all $i$, Hence, we can bound the RHS term within the probability.

$$\frac{x_{ti}^\top \theta^* - x_{ti}^\top \hat{\theta}_t}{\alpha_t \|x_{ti}\|_{V_t^{-1}}} \leq \frac{\alpha_t \|x_{ti}\|_{V_t^{-1}}}{\alpha_t \|x_{ti}\|_{V_t^{-1}}} = 1$$

Then, it follows that

$$\mathbb{P}\left(\max_j x_{ti}^\top \widetilde{\theta}_t^{(j)} > x_{ti}^\top \theta^* \mid \mathcal{F}_t\right) \geq 1 - (\mathbb{P}(Z \leq 1))^M. \tag{15}$$

Now, since $S_t = \arg\max_S \widetilde{R}_t(S)$, we have $\widetilde{R}_t(S_t) \geq \widetilde{R}_t(S_t^*)$. Then combining with Lemma 1, we can lower-bound the probability of having an expected revenue optimistic under the sampled parameter (the second inequality below).

$$
\begin{aligned}
\mathbb{P}\left(\widetilde{R}_t(S_t) > R_t(S_t^*, \theta_t^*) \mid \mathcal{F}_t\right) &\geq \mathbb{P}\left(\widetilde{R}_t(S_t^*) > R_t(S_t^*, \theta_t^*) \mid \mathcal{F}_t\right) \\
&\geq \mathbb{P}\left(\widetilde{u}_{ti} > x_{ti}^\top \theta^*, \forall i \in S_t^* \mid \mathcal{F}_t\right) \\
&= \mathbb{P}\left(\max_j x_{ti}^\top \widetilde{\theta}_t^{(j)} > x_{ti}^\top \theta^*, \forall i \in S_t^* \mid \mathcal{F}_t\right) \\
&\geq 1 - K\left(\mathbb{P}(Z \leq 1)\right)^M
\end{aligned}
$$

where the last inequality comes from Eq.(15) and the union bound. Using the anti-concentration inequality in Lemma 15, we have $\mathbb{P}(Z \leq 1) \leq 1 - \frac{1}{4\sqrt{e\pi}}$. Hence, it follows that

$$\mathbb{P}\left(\widetilde{R}_t(S_t) > R_t(S_t^*, \theta_t^*) \mid \mathcal{F}_t\right) \geq 1 - K\left(1 - \frac{1}{4\sqrt{e\pi}}\right)^M$$

$$\geq 1 - \left(1 - \frac{1}{4\sqrt{e\pi}}\right)$$

$$= \frac{1}{4\sqrt{e\pi}}$$

where the second inequality comes from our choice of $M = \lceil 1 - \frac{\log K}{\log\left(1 - 1/(4\sqrt{e\pi})\right)}\rceil$ which implies $\left(1 - \frac{1}{4\sqrt{e\pi}}\right)^M \leq \frac{1}{K}\left(1 - \frac{1}{4\sqrt{e\pi}}\right)$.

$\square$

### E.3 Proof of Lemma 12

*Proof.* The proof is inspired by the techniques used for Theorem 1 in [3]. First, we define $\widetilde{\Theta}_t$ the set of parameter samples for which the expected revenue concentrate appropriately to the expected revenue based on the MLE parameter. Also, we define the set of optimistic parameter samples $\widetilde{\Theta}_t^{\text{opt}}$ which coinciding with $\widetilde{\Theta}_t$.

$$\widetilde{\Theta}_t := \left\{\{\widetilde{\theta}_t^{(j)}\}_{j=1}^M : \widetilde{R}_t(S_t) - R_t(S_t, \hat{\theta}_t) \leq \beta_t \max_{i \in S_t} \|x_{ti}\|_{V_t^{-1}}\right\}$$

$$\widetilde{\Theta}_t^{\text{opt}} := \left\{\{\widetilde{\theta}_t^{(j)}\}_{j=1}^M : \widetilde{R}_t(S_t) > R_t(S_t^*, \theta_t^*)\right\} \cap \widetilde{\Theta}_t$$

Define the event $\mathcal{E}_t$ that both $x_{ti}^\top \hat{\theta}_t$ and $\widetilde{u}_{ti}$ are concentrated around their respective means.

$$\mathcal{E}_t = \{x_{ti}^\top \hat{\theta}_t - x_{ti}^\top \theta^* \leq \alpha_t \|x_{ti}\|_{V_t^{-1}}, \forall i\} \cap \{\widetilde{u}_{ti} - x_{ti}^\top \hat{\theta}_t \leq \beta_t \|x_{ti}\|_{V_t^{-1}}, \forall i\}.$$

For any $\widetilde{\theta}_t^{1:M} := \{\widetilde{\theta}_t^{(j)}\}_{j=1}^M \in \widetilde{\Theta}_t^{\text{opt}}$, we have

$$\left(R_t(S_t^*, \theta_t^*) - \widetilde{R}_t(S_t)\right)\mathbb{1}(\mathcal{E}_t) \leq \left(R_t(S_t^*, \theta_t^*) - \inf_{\theta_t^{1:M} \in \widetilde{\Theta}_t} \widetilde{R}_t(S_t, \theta_t^{1:M})\right)\mathbb{1}(\mathcal{E}_t)$$

where $\widetilde{R}_t(S_t, \theta_t^{1:M})$ is the optimistic expected revenue under the sampled parameters $\theta_t^{1:M}$. Then we can bound $R_t(S_t^*, \theta_t^*) - \widetilde{R}_t(S_t)$ by the expectation over any random choice $\widetilde{\theta}_t^{1:M} \in \widetilde{\Theta}_t^{\text{opt}}$

$$R_t(S_t^*, \theta_t^*) - \widetilde{R}_t(S_t) \leq \mathbb{E}\left[\left(\widetilde{R}_t(S_t) - \inf_{\theta_t^{1:M} \in \widetilde{\Theta}_t} \widetilde{R}_t(S_t, \theta_t^{1:M})\right)\mathbb{1}(\mathcal{E}_t) \mid \mathcal{F}_t, \widetilde{\theta}_t^{1:M} \in \widetilde{\Theta}_t^{\text{opt}}\right]$$

$$= \mathbb{E}\left[\sup_{\theta_t^{1:M} \in \widetilde{\Theta}_t} \left(\widetilde{R}_t(S_t) - \widetilde{R}_t(S_t, \theta_t^{1:M})\right)\mathbb{1}(\mathcal{E}_t) \mid \mathcal{F}_t, \widetilde{\theta}_t^{1:M} \in \widetilde{\Theta}_t^{\text{opt}}\right]$$

$$\leq \mathbb{E}\left[\sup_{\theta_t^{1:M} \in \widetilde{\Theta}_t} \max_{i \in S_t} \left|\widetilde{u}_{ti} - x_{ti}^\top \theta_t^{(j)}\right|\mathbb{1}(\mathcal{E}_t) \mid \mathcal{F}_t, \widetilde{\theta}_t^{1:M} \in \widetilde{\Theta}_t^{\text{opt}}\right]$$

$$\leq 2\beta_t \mathbb{E}\left[\max_{i \in S_t(\widetilde{\theta}_t^{1:M})} \|x_{ti}\|_{V_t^{-1}} \mid \mathcal{F}_t, \widetilde{\theta}_t^{1:M} \in \widetilde{\Theta}_t^{\text{opt}}, \mathcal{E}_t\right]\mathbb{P}(\mathcal{E}_t)$$

where the last inequality is from the definition of the set $\widetilde{\Theta}_t$ and $S_t(\widetilde{\theta}_t^{1:M})$ stands for the optimal assortment under the sampled parameters $\widetilde{\theta}_t^{1:M} = \{\widetilde{\theta}_t^{(j)}\}_{j=1}^M$.

From Lemma 2, we have $\mathbb{P}\left(\widetilde{R}_t(S_t) > R_t(S_t^*, \theta_t^*) \mid \mathcal{F}_t, \mathcal{E}_t\right) \geq \frac{1}{4\sqrt{e\pi}} =: \widetilde{p}$. Therefore it follows that

$$
\begin{aligned}
\mathbb{P}\left(\widetilde{\theta}_t^{1:M} \in \widetilde{\Theta}_t^{\text{opt}} \mid \mathcal{F}_t, \mathcal{E}_t\right) &= \mathbb{P}\left(\widetilde{R}_t(S_t) > R_t(S_t^*, \theta_t^*) \text{ and } \widetilde{\theta}_t^{1:M} \in \widetilde{\Theta}_t, \mathcal{E}_t\right) \\
&\geq \mathbb{P}\left(\widetilde{R}_t(S_t) > R_t(S_t^*, \theta_t^*) \mid \mathcal{F}_t, \mathcal{E}_t\right) - \mathbb{P}\left(\widetilde{\theta}_t^{1:M} \notin \widetilde{\Theta}_t, \mathcal{E}_t\right) \\
&\geq \widetilde{p} - \mathcal{O}(t^{-1}) \\
&\geq \widetilde{p}/2.
\end{aligned}
$$

Now, note that we can write

$$
\begin{aligned}
\mathbb{E}\left[\max_{i \in S_t(\widetilde{\theta}_t^{1:M})} \|x_{ti}\|_{V_t^{-1}} \mid \mathcal{F}_t, \mathcal{E}_t\right] &\geq \mathbb{E}\left[\max_{i \in S_t(\widetilde{\theta}_t^{1:M})} \|x_{ti}\|_{V_t^{-1}} \mid \mathcal{F}_t, \widetilde{\theta}_t^{1:M} \in \widetilde{\Theta}_t^{\text{opt}}, \mathcal{E}_t\right] \mathbb{P}\left(\widetilde{\theta}_t^{1:M} \in \widetilde{\Theta}_t^{\text{opt}} \mid \mathcal{F}_t, \mathcal{E}_t\right) \\
&\geq \mathbb{E}\left[\max_{i \in S_t(\widetilde{\theta}_t^{1:M})} \|x_{ti}\|_{V_t^{-1}} \mid \mathcal{F}_t, \widetilde{\theta}_t^{1:M} \in \widetilde{\Theta}_t^{\text{opt}}, \mathcal{E}_t\right] \cdot \widetilde{p}/2
\end{aligned}
$$

Therefore, combining the results, we have

$$
\begin{aligned}
R_t(S_t^*, \theta_t^*) - \widetilde{R}_t(S_t) &\leq 2\beta_t \mathbb{E}\left[\max_{i \in S_t(\widetilde{\theta}_t^{1:M})} \|x_{ti}\|_{V_t^{-1}} \mid \mathcal{F}_t, \widetilde{\theta}_t^{1:M} \in \widetilde{\Theta}_t^{\text{opt}}, \mathcal{E}_t\right] \mathbb{P}(\mathcal{E}_t) \\
&\leq \frac{4\beta_t}{\widetilde{p}} \mathbb{E}\left[\max_{i \in S_t(\widetilde{\theta}_t^{1:M})} \|x_{ti}\|_{V_t^{-1}} \mid \mathcal{F}_t, \mathcal{E}_t\right] \mathbb{P}(\mathcal{E}_t) \\
&\leq \frac{4\beta_t}{\widetilde{p}} \mathbb{E}\left[\max_{i \in S_t(\widetilde{\theta}_t^{1:M})} \|x_{ti}\|_{V_t^{-1}} \mid \mathcal{F}_t\right].
\end{aligned}
$$

Summing over all $t$ and taking the failure event into consideration, we have

$$
\sum_{t=1}^T \left(R_t(S_t^*, \theta_t^*) - \widetilde{R}_t(S_t)\right) \leq \sum_{t=1}^T \frac{4\beta_t}{\widetilde{p}} \mathbb{E}\left[\max_{i \in S_t(\widetilde{\theta}_t^{1:M})} \|x_{ti}\|_{V_t^{-1}} \mid \mathcal{F}_t\right].
$$

Here, the summation on the RHS contains an expectation, so we cannot directly apply Lemma 6. Instead, we use Lemma 14 to bound the sum of the expectations

$$
\sum_{t=1}^T \mathbb{E}[R_t(S_t^*, \theta_t^*) - \widetilde{R}_t(S_t)] \leq \sum_{t=1}^T \frac{4\beta_t}{\widetilde{p}}\left(\sqrt{2dT \log\left(1 + \frac{TK}{d\lambda}\right)} + \sqrt{\frac{8T}{\lambda} \log 2T}\right) + \mathcal{O}(1).
$$

$\square$

**Lemma 14.** *If $\lambda_{\min}(V_t) \geq \lambda$, then with probability $1 - \mathcal{O}(T^{-1})$ we have*

$$
\sum_{t=1}^T \mathbb{E}\left[\max_{i \in S_t(\widetilde{\theta}_t^{1:M})} \|x_{ti}\|_{V_t^{-1}} \mid \mathcal{F}_t\right] \leq \sqrt{2dT \log\left(1 + \frac{TK}{d\lambda}\right)} + \sqrt{\frac{8T}{\lambda} \log 2T}.
$$

*Proof.* We rewrite the summation as follows.

$$
\begin{aligned}
&\sum_{t=1}^T \mathbb{E}\left[\max_{i \in S_t(\widetilde{\theta}_t^{1:M})} \|x_{ti}\|_{V_t^{-1}} \mid \mathcal{F}_t\right] \\
&= \sum_{t=1}^T \max_{i \in S_t} \|x_{ti}\|_{V_t^{-1}} + \sum_{t=1}^T \left(\mathbb{E}\left[\max_{i \in S_t(\widetilde{\theta}_t^{1:M})} \|x_{ti}\|_{V_t^{-1}} \mid \mathcal{F}_t\right] - \max_{i \in S_t} \|x_{ti}\|_{V_t^{-1}}\right)
\end{aligned} \tag{16}
$$

The first summation can be bounded by using Lemma 6 and Cauchy-Schwarz inequality.

$$
\sum_{t=1}^T \max_{i \in S_t} \|x_{ti}\|_{V_t^{-1}} \leq \sqrt{T \sum_{t=1}^T \max_{i \in S_t} \|x_{ti}\|_{V_t^{-1}}^2} \leq \sqrt{2dT \log\left(1 + \frac{TK}{d\lambda}\right)} \tag{17}
$$

For the second summation in Eq.(16), we can apply Azuma-Hoeffding inequality (Lemma 16). Note that the second summation is a martingale by construction. Also recall that $\max_{i \in S_t} \|x_{ti}\| \leq 1$ for all $t$, hence we have

$$\mathbb{E}\left[\max_{i \in S_t(\widetilde{\theta}_t^{1:M})} \|x_{ti}\|_{V_t^{-1}} \mid \mathcal{F}_t\right] - \max_{i \in S_t} \|x_{ti}\|_{V_t^{-1}} \leq \frac{2}{\lambda_{\min}(V_t)} \leq \frac{2}{\lambda_{\min}(V)} = \frac{2}{\lambda}.$$

Therefore, $\frac{2}{\lambda}$ is an upper-bound for each element in the second summation. Now applying Azuma-Hoeffding inequality, we have

$$\sum_{t=1}^{T} \left(\mathbb{E}\left[\max_{i \in S_t(\widetilde{\theta}_t^{1:M})} \|x_{ti}\|_{V_t^{-1}} \mid \mathcal{F}_t\right] - \max_{i \in S_t} \|x_{ti}\|_{V_t^{-1}}\right) \leq \sqrt{\frac{8T}{\lambda} \log 2T} \tag{18}$$

with probability $1 - \mathcal{O}(T^{-1})$. Combining Eq.(17) and Eq.(18), we have the result. $\qquad \square$

### E.4 Other Lemmas

The following lemma is used to derive the concentration and anti-concentration inequalities for Gaussian random variables.

**Lemma 15** (Abramowitz and Stegun 4). *For a Gaussian random variable $Z$ with mean $\mu$ and variance $\sigma^2$, for any $z \geq 1$,*

$$\frac{1}{2\sqrt{\pi}z} e^{-z^2/2} \leq \mathbb{P}\left(|Z - \mu| > z\sigma\right) \leq \frac{1}{\sqrt{\pi}z} e^{-z^2/2}. \tag{19}$$

**Lemma 16** (Azuma-Hoeffding inequality). *If a super-martingale $(Y_t; t \geq 0)$ corresponding to filtration $\mathcal{F}_t$, satisfies $|Y_t - Y_{t-1}| \leq c_t$ for some constant $c_t$, for all $t = 1, ..., T$, then for any $a \geq 0$,*

$$\mathbb{P}(Y_T - Y_0 \geq a) \leq 2e^{-\frac{a^2}{2\sum_{t=1}^{T} c_t^2}}$$

## F    Guarantees for Random Initialization

As we discussed briefly in Section A, TS-MNL can start with the random initialization phase where the agent randomly chooses an assortment $S_t$ instead of using regularization in the parameter estimation. However, the length of the initialization $T_0$ needs to be specified in order to ensure a unique solution of MLE for a theoretical guarantee.

We maintain $V_{T_0} = \sum_{\tau=1}^{T_0} \sum_{i \in S_\tau} x_{\tau i} x_{\tau i}^\top$ while choosing assortments randomly during the random initialization. The initialization duration $T_0$ is chosen to ensure that $\lambda_{\min}(V_{T_0})$ is large enough so that $V_{T_0}$ is invertible. The following proposition allows us to find such $T_0$.

**Proposition 1.** *Let $x_{\tau i}$ be drawn i.i.d. from some distribution with $\|x_{\tau i}\| \leq 1$ and $\mathbb{E}[x_{\tau i} x_{\tau i}^\top] \geq \sigma_0$. Define $V_{T_0} = \sum_{\tau=1}^{T_0} \sum_{i \in S_\tau} x_{\tau i} x_{\tau i}^\top$, where $T_0$ is the length of random initialization. Suppose we run a random initialization with assortment size $K$ for duration $T_0$ which satisfies*

$$T_0 \geq \frac{1}{K}\left(\frac{C_1 \sqrt{d} + C_2 \sqrt{\log T}}{\sigma_0}\right)^2 + \frac{2B}{K\sigma_0}$$

*for some positive, universal constants $C_1$ and $C_2$. Then, $\lambda_{\min}(V_{T_0}) \geq B$ with probability at least $1 - T^{-1}$.*

The proposition is the adaptation of Proposition 1 in [30], modified for our multinomial setting. If we use $B = K$, then the proposition implies that we can have $\lambda_{\min}(V_{T_0}) \geq K$ with a high probability if we run the initialization for $\mathcal{O}(\sigma_0^{-2}(d + \log T))$ rounds. Similar to [23] and [30], the i.i.d. assumption on the context $x_{ti}$ may be only needed to ensure that $V_\tau$ is invertible at the end of the initialization phase. Hence, after the initialization, $x_{ti}$ can even be chosen adversarially as long as $\|x_{ti}\|$ is bounded.

## G  Numerical Study Details

### G.1  Synthetic Experiments

For synthetic experiments, we first sample feature vectors $x_i$ for each $i \in [N]$ in $d-1$ dimension with each entry from the standard Gaussian distribution. We then normalize this vectors and add an extra dimension with constant 1 for the intercept and divide by $\sqrt{2}$ so that the $\ell_2$ norm of feature vectors is bounded, i.e., $\|x_i\| \leq 1$. Similarly, we sample the parameter $\theta^*$ from the $d$-dimensional standard multivariate Gaussian distribution but without the normalization. For each experimental instance, we draw new samples of $\{x_i\}$ and $\theta^*$.

In Figure 1, we only showed the performance of the UCB algorithm proposed in [15] for $N = 256$. The UCB algorithm proposed in [15] constructs confidence bounds for each of ($N$ choose $K$) assortments (as discussed in Section 2), the evaluation on a larger $N$ causes a significant computational burden; hence we had to keep $N$ at a reasonable size for evaluating the UCB method. In fact, even with $N = 256$, we could not use the original version of the UCB method in [15] due to the computational complexity. We instead use a greedy heuristic for solving the combinatorial optimization proposed as an alternative efficient approximation (see Algorithm 4 in [15]) although it does not have rigorous guarantees. However, it is important to note that even with such computational compromises for the UCB method, our TS methods still have better computational efficiency as well as superior performances on the statistical efficiency. Note that our proposed methods do not suffer from this issue and can be evaluated with a much larger $N$ which is shown in the experiments in Figure 2 as well as the MovieLens experiment (with $N = 1000$) in Figure 1.

The left plot in Figure 2 shows the evaluations of TS-MNL with optimistic sampling with varying feature dimensions. The reported results are averaged over 40 independent instances. The results show that the performance of our algorithm is still attractive even with an increase in the feature dimension, which shows a better scalability in $d$ than the theoretical guarantees, at most $d^{3/2}$ dependence on the worst-case regret bound.

Furthermore, the experiments in the right plot of Figure 2 show that even when the number of total items $N$ increases, the empirical performances of our proposed algorithms remain the same as the performance in Figure 1 and are not hindered by such an increase in $N$. This observation is consistent with our established theoretical results and supports the claim that our methods can be used and effective for problem instances with very large $N$ — as long as the combinatorial optimization step can be efficiently computed.

Figure 2: Experiments on varying feature dimensions and with an increased number of items

### G.2  Experiments with MovieLens Dataset

**Dataset.** MovieLens datasets[3] contain the ratings of users for movies from the MovieLens website. The datasets come in different sizes and we use "MovieLens 20M" for our experiments. This dataset contains 20 million ratings of $2.7 \times 10^4$ movies by $1.38 \times 10^5$ users.

**Feature extraction.**  We follow the experimental setup of [31].  For our experiments, we use $N = 1000$ movies with most ratings and $1.1 \times 10^3$ user with the most number of ratings.  We

randomly split the user set into two parts $A_1$ and $A_2$ with $|A_1| = 100$ and $|A_2| = 1000$. Then we use the matrix of the movie rating for users in $A_1$ to extract feature vectors with $d = 5$. Note that the MovieLens dataset does not come with movie or user features — it only contains ratings of the movies by the user as a matrix, which we denote as $W$. Hence, we construct features for our experiments using the collaborative filtering approach. We derive the features of movies using low-rank matrix factorization.

Splitting the user set into two parts $A_1$ and $A_2$ means dividing the rows of the matrix $W$ into two matrices: one with 100 rows corresponding to $A_1$ and the other with 1000 rows corresponding to $A_2$. We define training matrix $W_{\text{train}} \in \mathbb{R}^{|A_1| \times N}$ and test matrix $W_{\text{test}} \in \mathbb{R}^{|A_2| \times N}$ corresponding to user sets $A_1$ and $A_2$ respectively. We use $W_{\text{train}}$ to learn the features of items and $W_{\text{test}}$ to evaluate our learning algorithms.

Let $W_{\text{train}} \approx U \Sigma V^\top$ be rank-$d$ truncated SVD of $W_{\text{train}}$, where $U \in \mathbb{R}^{|A_1| \times d}$, $\Sigma \in \mathbb{R}^{d \times d}$, and $V \in \mathbb{R}^{N \times d}$. Then the features of movies are the rows of $V\Sigma$. Note that the matrix $V$ in the section is defined within this experimental setup only and is different from the gram matrix $V_t$ used in the regret analysis or in the algorithm. We overload this term for the sake of consistency with terms typically used in matrix factorization literature.

**Offline regression.** We use the extracted features of movies, i.e., rows of $\bar{V}\Sigma$, and the mean score of each of the movies considered. The true parameter $\theta^*$ is computed by solving the linear system of $N$ with respect to the rating matrix of $W_{\text{test}}$.

**Evaluations.** Once we extract the features and learn $\theta^*$, we perform online evaluations. We set $\lambda$ to be the same as the feature dimension $d$. Note that the dataset does not contain separate revenue information for different movies. Hence we assume that the revenue parameter is uniform across all movies, i.e. each user choice/click is equally weighted. Therefore, the combinatorial optimization step reduces to sorting items according to estimated utilities and choosing top $K$ movies. The evaluation results show that two variants of TS-MNL are effective. In particular, TS-MNL with optimistic sampling shows more attractive performances in these sample results.