[Reviews · NeurIPS 2019]

Reviewer 1



The experimental section is tepid. As such this paper needs to be evaluated primarily on theoretical merit. Under this criterion I find the paper easily exceeds the standards for publication.

Reviewer 2



I think overall this paper is a good contribution. The Bayesian and the frequentist analysis for the TS-MNL algorithm is novel. Especially the frequentist analysis builds on recently proposed proof techniques in [2,4] (for linear contextual bandits) and extends it to the MNL problem. The frequentist regret bounds matches the corresponding bounds in the linear case. I have a few concerns and it would be great if the authors can correct/respond to them: 1. Concerns: Lines 142-144 is confusingly written. y_{t,i} is supposed to belong in {0,1} as noted before. It is better to write that the y vector is sampled from the multi-nomial logit distribution given by equation below line 140, rather the sub-gaussianity based description where it seems like y_{t,i} can take values in [0,1], even though it is technically correct. Assumption 1 actually says that x_{t,i} is actually i.i.d. This should be clarified before in the paper, as there are multiple mentions of x_{t,i} varying over time. It should be clarified that the distribution is i.i.d. Line 4 in the algorithm: I think the monte-carlo sampling scheme should be discussed in more details in the body of the paper. Also, how does this approximate sampling the theoretical or practical regret performance of the algorithm, needs to be discussed. The exact details of the markov chains construction are missing from the appendix as well. Line 201: There is no needs to use the term oracle if a polynomial time algorithm exists to solve the problem exactly. You can just refer to the algorithm (I am just commenting on the usage of the term oracle). It would also be appreciated if the computational complexity of the said algorithm is revealed in the discussion. Add a list of differences from [6] earlier in the paper. A little more details about the experimental setup is needed for the main body of the paper. For example, which algorithm was used as the optimization algorithm for line 5 in the algorithm. The real experiments are essentially semi-synthetic. A model is learnt from the actual dataset and then that model is used as the ground truth parameter. Can you think of a better evaluation method? Typos: Line 24 -- exploitation-exploitation trade-off -> exploration-exploitation trade-off Line 24-25 -- there are epoch/epsilon greedy strategies as well Line 136: and and -> and

Reviewer 3



Overall, the paper is fine: setting, proposed approaches, theoretical analysis, and clarity. Still it may be improved in few ways. The main improvement would regard the motivation of the paper. Typically * the paper should give insight on why the best assortment selection with current assumption wrt. the reward distribution is different from the best assumption with cascading click model, or position-based click models; * the paper claims frugality of the proposed approach wrt. computation time (compared to UCB based approaches). That claim could be enforced by providing the computation time measured in experiments. It would also be enforced by a theoretical analysis of computation time. * current experiments are on simulated data (derived from real-world data). Experiments on real-world data with assortments would be more convincing. It's also unusual to have bibliographic citations linking to arXiv version of each paper without reference to the conference/journal where the paper was accepted. To name one, paper [39] has been accepted at UAI'16. ____________________________ POST REBUTTAL --- I thank the authors for their thorough answer. I agree with the authors regarding the originality and the soundness of assortment optimization as opposed to cascading/position-based model (let's denote them "independent optimization models") for multiple recommendation setting. Still, I think THE PAPER should include as much insights as possible to demonstrate the potentiality of assortment optimization: * some simple examples where assortment and independent optimization are selecting two different assortments (with substituable items, obviously), * some real-world datasets/analysis demonstrating the benefit of such modelization, * an empirical comparison to state of the art regarding independent optimization models. Such comparaison is even more required given that several papers with independent modelization were published in Machine Learning avenues. By the way, I would prefer the semi-synthetic data to be derived from real-world datasets with assortment properties. There are publicly available "baskets" datasets, I wonder if they may provide the required starting data. Finally, I acknowledge that appart from paper [39], arXiv citations concern papers which are still unpublished. Having five of them cited remains surprising.

[Author Response · NeurIPS 2019]

We thank the reviewers for their time in reviewing our paper and their constructive feedback. We emphasize that the dynamic assortment selection problem we address in this paper is a fundamental and general problem – combining with contextual information and an easy extension to position-based offering (see the response to Reviewer 3) makes our setting one of the most common forms of recommendations one may face in practice. We propose the methods which are both theoretically sound and practical. The responses to specific questions/issues raised by each reviewer are presented below.

## Response to Reviewer 1

The parameter $\theta^*$ is modeled from the entire data considered and then in each round the MNL model is simulated with this $\theta^*$. We will make it more clear in the paper.

We are currently working on the experiments with model mis-specification where the true model is not the MNL model. We will include the results in the paper. We appreciate your input.

## Response to Reviewer 2

We agree with the suggested corrections and appreciate your feedback. On Assumption 1, while it states $x_{t,i}$ is i.i.d., we emphasize that the i.i.d. assumption on $x_{t,i}$ is only required during the initialization phase to ensure the invertibility of $V_{T_0}$ (in order to have a unique solution of MLE as mentioned in Appendix B.1). After the initialization, $x_{t,i}$ can even be chosen adversarially as long as $\|x_{t,i}\|$ is bounded. Now, we note that in practice the same can be achieved by introducing regularization, but for a better tractability of the analysis we chose random initialization along with the i.i.d. assumption (at least for the initialization). We will make it more clear in the paper.

As for the experiments, we will include more details about the experimental setup in the main body of the paper if the space permits. On the optimization step of the experiment, MovieLens dataset does not contain different revenue value for movies. Therefore, it is equivalent to having the unit revenue for all movies. Hence, the optimization step reduces to a sorting task based on estimated utility. We will include additional experimental results where we use synthetic data which contains the (synthetic) revenue parameter for each item. For that, we use the LP solution proposed in [17].

The experiments are indeed semi-synthetic. The interactive aspect of bandit problems (not just our MNL bandit) makes it notoriously difficult to evaluate in real-world settings unless one performs a field experiment. That is why most bandit papers perform evaluations with completely synthetic data. As mentioned in the previous paragraph, we will include additional experiment results with synthetically generated data.

## Response to Reviewer 3

Our work (as well as previous work in MNL bandit) is distinct from other combinatorial bandit problems such as cascading bandits and semi-bandits. In typical cascading or semi-bandit settings, the mapping from the item context to the user feedback is independent of other items in an offered set. On the other hand, MNL choice feedback is a function of entire assortment which makes our analysis much more challenging.

Regarding position-based offering, we can easily incorporate display position effect within the assortment by including a categorical variable indicating the display position in the context vector. Hence, we also estimate parameters corresponding to each display position. We can show that our algorithms are able to still use the LP solution [17] for this position-dependent extension of the combinatorial optimization problem. Note that this extension is different from previous position-based click models in which the user feedback is typically a function of an item context independent of other items in an offered set. This position-based extension of our framework still takes into account the substitution effect within the assortment.

We argue that in practice the assortment based offering (with or without display position effect) is the most prevalent form of recommendations in online retailing (e.g. Amazon, Walmart), streaming services (e.g. Netflix), news websites, and many more – in fact, one rarely faces single-item offering (typical bandit setting) or item-wise cascading offering in those common applications. Furthermore, when an assortment (a set of items) is offered, there is often a substitution effect among the items, which many other combinatorial bandit models do not address. We appreciate your feedback and the chance to re-emphasize our motivation.

We will add additional experimental results on the computation time. Regarding the evaluation on real-world data, (as mentioned in the response to Reviewer 2) the interactive aspect of bandit problems (not just our MNL bandit) makes it notoriously difficult to evaluate in real-world settings unless one performs a field experiment.

Thank you for pointing out the citation mistake. We replaced the arXiv version of [39] with its UAI 2016 publication. We double-checked the other arXiv papers which we cited in our paper and confirmed that they did not appear in any previous proceedings or journals.

[Meta-Review · NeurIPS 2019]

Reviewers were generally positive. A well prepared article that analyzes the practical Thompson Sampling heuristic for the prescribed problem. The problem statement and techniques do not appear ground-breaking, but effective for the task.